# Explore to Generalize in Zero-Shot RL

**Ev Zisselman**,[*] **Itai Lavie, Daniel Soudry, Aviv Tamar**
Technion – Israel Institute of Technology

## Abstract

We study zero-shot generalization in reinforcement learning—optimizing a policy on a set of training tasks to perform well on a similar but unseen test task. To mitigate overfitting, previous work explored different notions of invariance to the task. However, on problems such as the ProcGen Maze, an adequate solution that is invariant to the task visualization does not exist, and therefore invariance-based approaches fail. Our insight is that learning a policy that effectively *explores* the domain is harder to memorize than a policy that maximizes reward for a specific task, and therefore we expect such learned behavior to generalize well; we indeed demonstrate this empirically on several domains that are difficult for invariance-based approaches. Our *Explore to Generalize* algorithm (ExpGen) builds on this insight: we train an additional ensemble of agents that optimize reward. At test time, either the ensemble agrees on an action, and we generalize well, or we take exploratory actions, which generalize well and drive us to a novel part of the state space, where the ensemble may potentially agree again. We show that our approach is the state-of-the-art on tasks of the ProcGen challenge that have thus far eluded effective generalization, yielding a success rate of $83\%$ on the Maze task and $74\%$ on Heist with 200 training levels. ExpGen can also be combined with an invariance based approach to gain the best of both worlds, setting new state-of-the-art results on ProcGen. Code available at `https://github.com/EvZissel/expgen`.

## 1 Introduction

Recent developments in reinforcement learning (RL) led to algorithms that surpass human experts in a broad range of tasks [Mnih et al., 2015, Vinyals et al., 2019, Schrittwieser et al., 2020, Wurman et al., 2022]. In most cases, the RL agent is tested on the same task it was trained on, and is not guaranteed to perform well on unseen tasks. In zero-shot generalization for RL (ZSG-RL), however, the goal is to train an agent on training domains to act optimally in a new, previously unseen test environment [Kirk et al., 2021]. A standard evaluation suite for ZSG-RL is the ProcGen benchmark [Cobbe et al., 2020], containing 16 games, each with levels that are procedurally generated to vary in visual properties (e.g., color of agents in BigFish, Fig. 1a, or background image in Jumper, Fig. 1c) and dynamics (e.g., wall positions in Maze, Fig. 1d, and key positions in Heist, Fig. 1e).

Previous studies focused on identifying various *invariance properties* in the tasks, and designing corresponding *invariant policies*, through an assortment of regularization and augmentation techniques [Igl et al., 2019, Cobbe et al., 2019, Wang et al., 2020, Lee et al., 2019a, Raileanu et al., 2021, Raileanu and Fergus, 2021, Cobbe et al., 2021, Sonar et al., 2021, Bertran et al., 2020, Li et al., 2021]. For example, a policy that is invariant to the color of agents is likely to generalize well in BigFish. More intricate invariances include the order of observations in a trajectory [Raileanu and Fergus, 2021], and the length of a trajectory, as reflected in the value function [Raileanu and Fergus, 2021].

---

[*]Correspondence E-mail: `ev_zis@campus.technion.ac.il`

37th Conference on Neural Information Processing Systems (NeurIPS 2023).

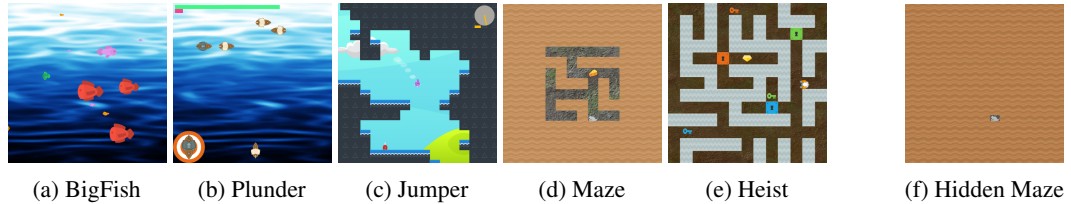

|  (a) BigFish | (b) Plunder | (c) Jumper | (d) Maze | (e) Heist | (f) Hidden Maze |

Figure 1: (a),(b),(c),(d) and (e) displays screenshot of ProcGen games. (f) Imaginary maze with goal and walls removed (see text for explanation).

Can ZSG-RL be reduced to only finding invariant policies? As a counter-argument, consider the following thought experiment[2]. Imagine Maze, but with the walls and goal hidden in the observation (Fig. 1f). Arguably, this is the most task-invariant observation possible, such that a solution can still be obtained in a reasonable time. An agent with memory can be trained to optimally solve all training tasks: figuring out wall positions by trying to move ahead and observing the resulting motion, and identifying based on its movement history in which training maze it is currently in. Obviously, such a strategy will not generalize to test mazes. Indeed, as depicted in Figure 2, performance in tasks like Maze and Heist, where the strategy for solving any particular training task must be *indicative* of that task, has largely not improved by methods based on invariance (e.g. UCB-DrAC and IDAAC).

Interestingly, decent zero-shot generalization can be obtained even without a policy that generalizes well. As described by Ghosh et al. [2021], an agent can overcome test-time errors in its policy by treating the perfect policy as an *unobserved* variable. The resulting decision making problem, termed the *epistemic POMDP*, may require some exploration at test time to resolve uncertainty. Ghosh et al. [2021] further proposed the LEEP algorithm based on this principle, which trains an ensemble of agents and essentially chooses randomly between the members when the ensemble does not agree, and was the first method to present substantial generalization improvement on Maze.

In this work, we follow the epistemic POMDP idea, but ask: *how to improve exploration at test time?* Our approach is based on a novel discovery: when we train an agent to *explore* the training domains using a maximum entropy objective [Hazan et al., 2019, Mutti et al., 2021], we observe that the learned exploration behavior generalizes surprisingly well—much better than the generalization attained when training the agent to maximize reward. Intuitively, this can be explained by the fact that reward is a strong signal that leads to a specific behavior that the agent can 'memorize' during training, while exploration is naturally more varied, making it harder to memorize and overfit.

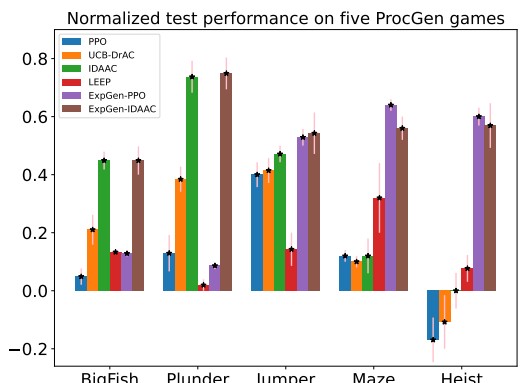

Figure 2: Normalized test Performance for Exp-Gen, LEEP, IDAAC, DAAC, and PPO, on five ProcGen games. ExpGen shows state-of-the-art performance on test levels of Maze, Heist and Jumper; games that are notoriously challenging for other leading approaches. The scores are normalized as proposed by [Cobbe et al., 2020].

Exploration by itself, however, is not useful for solving new tasks. Our algorithm, *Explore to Generalize* (ExpGen), additionally trains an ensemble of reward-seeking agents. At test time, either the ensemble agrees on an action, and we generalize well, or we take exploratory actions *using the exploration policy*, which we demonstrate to generalize, and drive us to a novel part of the state space, where the ensemble may potentially agree again.

ExpGen is simple to implement, and can be used with any reward maximizing RL algorithm. Combined with vanilla PPO, ExpGen significantly improves the state-of-the-art (SOTA) on several ProcGen games for which previous methods fail (see Fig. 2). ExpGen also significantly improves

---

[2]We validated this experiment empirically, using a recurrent policy on Maze with 128 training tasks, observing 85% success rate on training domains, and test success similar to a random policy (see Appendix A).

upon LEEP, due to its effective test-time exploration strategy. For example, on Maze with 200 training levels, our method obtains $83\%$ success on test tasks, whereas the previous state-of-the-art achieved $66\%$. When combined with IDAAC [Raileanu and Fergus, 2021], the leading invariance-based algorithm, ExpGen achieves state-of-the-art performance on the full ProcGen suite (the full results are provided in Appendix D).

## 2    Related Work

**Generalization in RL**    The recent survey by Kirk et al. [2021] provides an extensive review of generalization in RL; here, we provide a brief overview. One approach to generalization is by artificially increasing the number of training tasks, using either procedural generation [Cobbe et al., 2019, 2020], or augmentations [Kostrikov et al., 2020, Ye et al., 2020, Lee et al., 2019a, Raileanu et al., 2021], task interpolation [Yao et al., 2021] or various regularization technique, such as dropout [Igl et al., 2020] and batch normalization [Farebrother et al., 2018, Igl et al., 2020]. Leading approaches, namely IDAAC [Raileanu and Fergus, 2021] and PPG [Cobbe et al., 2021], investigate the advantages of decoupling policy and value functions for generalization, whereas Jiang et al. [2021] propose automatic curriculum learning of levels.

A different approach is to add inductive bias to the neural network policy or learning algorithm. Approaches such as Tamar et al. [2016], Vlastelica et al. [2021], Boutilier et al. [2020] embed a differentiable planner or learning algorithm into the neural network. Other methods [Kansky et al., 2017, Toyer et al., 2018, Rivlin et al., 2020] combine learning with classical graph planning to generalize across various planning domains. These approaches require some knowledge about the problem structure (e.g., a relevant planning algorithm), while our approach does not require any task-specific knowledge. Another line of work aims to learn policies or features that are invariant across the different training tasks [Sonar et al., 2021, Bertran et al., 2020, Li et al., 2021, Igl et al., 2019, Stooke et al., 2021, Mazoure et al., 2020] and are thus more robust to sensory variations.

Using an ensemble to direct exploration to unknown areas of the state space was proposed in the model-based TEXPLORE algorithm of Hester and Stone [2013], where an ensemble of transition models was averaged to induce exploratory actions when the models differ in their prediction. Observing that exploration can help with zero-shot generalization, the model-free LEEP algorithm by Ghosh et al. [2021] is most relevant to our work. LEEP trains an ensemble of policies, each on a separate subset of the training environment, with a loss function that encourages agreement between the ensemble members. Effectively, the KL loss in LEEP encourages random actions when the agents of the ensemble do not agree, which is related to our method. However, random actions can be significantly less effective in exploring a domain than a policy that is explicitly trained to explore, such as a maximum-entropy policy. Consequentially, we observe that our approach leads to significantly better performance at test time.

**State Space Maximum Entropy Exploration**    Maximum entropy exploration (maxEnt, Hazan et al. 2019, Mutti et al. 2021) is an unsupervised learning framework that trains policies that maximize the entropy of their state-visitation frequency, leading to a behavior that continuously explores the environment state space. Recently, maximum entropy policies have gained attention in RL [Liu and Abbeel, 2021b,a, Yarats et al., 2021, Seo et al., 2021, Hazan et al., 2019, Mutti et al., 2021] mainly in the context of unsupervised pre-training. In that setting, the agent is allowed to train for a long period without access to environment rewards, and only during test the agent gets exposed to the reward signal and performs a limited fine-tuning adaptation learning. Importantly, these works expose the agent to the same environments during pre-training and test phases, with the only distinction being the lack of extrinsic reward during pre-training. To the best of our knowledge, our observation that maxEnt policies generalize well in the zero-shot setting is novel.

## 3    Problem Setting and Background

We describe our problem setting and provide background on maxEnt exploration.

**Reinforcement Learning (RL)**

In Reinforcement Learning an agent interacts with an unknown, stochastic environment and collects rewards. This is modeled by a Partially Observed Markov Decision Process (POMDP) [Bertsekas, 2012], which is the tuple $M = (S, A, O, P_{init}, P, \Sigma, r, \gamma)$, where $S \in \mathbb{R}^{|S|}$ and $A \in \mathbb{R}^{|A|}$ are the state and actions spaces, $O$ is the observation space, $P_{init}$ is an initial state distribution, $P$ is the transition kernel, $\Sigma$ is the observation function, $r : S \times A \to \mathbb{R}$ is the reward function, and $\gamma \in [0, 1)$ is the discount factor. The agent starts from initial state $s_0 \sim P_{init}$ and at time $t$ performs an action $a_t$ on the environment that yields a reward $r_t = r(s_t, a_t)$, and an observation $o_t = \Sigma(s_t, a_t) \in O$. Consequently, the environment transitions into the next state according to $s_{t+1} \sim P(\cdot|s_t, a_t)$. Let the history at time $t$ be $h_t = \{o_0, a_0, r_0, o_1, a_1, r_1 \dots, o_t\}$, the sequence of observations, actions and rewards. The agent's next action is outlined by a policy $\pi$, which is a stochastic mapping from the history to an action probability $\pi(a|h_t) = P(a_t = a|h_t)$. In our formulation, a history-dependent policy (and not a Markov policy) is required both due to partially observed states, epistemic uncertainty [Ghosh et al., 2021], and also for optimal maxEnt exploration [Mutti et al., 2022].

**Zero-Shot Generalization for RL**

We assume a prior distribution over POMDPs $P(M)$, defined over some space of POMDPs. For a given POMDP, an optimal policy maximizes the expected discounted return $\mathbb{E}_{\pi,M}[\sum_{t=0}^{\infty} \gamma^t r(s_t, a_t)]$, where the expectation is taken over the policy $\pi(h_t)$, and the state transition probability $s_t \sim P$ of POMDP $M$. Our generalization objective in this work is to maximize the discounted cumulative reward taken *in expectation over the POMDP prior*, also termed the *population risk*:

$$\mathcal{R}_{pop}(\pi) = \mathbb{E}_{M \sim P(M)} \left[ \mathbb{E}_{\pi,M} \left[ \sum_{t=0}^{\infty} \gamma^t r(s_t, a_t) \right] \right]. \tag{1}$$

Seeking a policy that performs well in expectation over any POMDP from the prior corresponds to zero-shot generalization.

We assume access to $N$ training POMDPs $M_1, \dots, M_N$ sampled from the prior, $M_i \sim P(M)$. Our goal is to use $M_1, \dots, M_N$ to learn a policy that performs well on objective 1. A common approach is to optimize the *empirical risk* objective:

$$\mathcal{R}_{emp}(\pi) = \frac{1}{N} \sum_{i=1}^{N} \mathbb{E}_{\pi,M_i} \left[ \sum_{t=0}^{\infty} \gamma^t r(s_t, a_t) \right] = \mathbb{E}_{M \sim \hat{P}(M)} \left[ \mathbb{E}_{\pi,M} \left[ \sum_{t=0}^{\infty} \gamma^t r(s_t, a_t) \right] \right], \tag{2}$$

where the empirical POMDP distribution can be different from the true distribution, i.e. $\hat{P}(M) \neq P(M)$. In general, a policy that optimizes the empirical risk (Eq. 2) may perform poorly on the population risk (Eq. 1)—this is known as overfitting in statistical learning theory [Shalev-Shwartz and Ben-David, 2014], and has been analyzed recently also for RL [Tamar et al., 2022].

**Maximum Entropy Exploration**

In the following we provide the definitions for the state distribution and the maximum entropy exploration objective. For simplicity, we discuss MDPs—the fully observed special case of POMDPs where $O = S$, and $\Sigma(s, a) = s$.

A policy $\pi$, through its interaction with an MDP, induces a $t$-step state distribution $d_{t,\pi}(s) = p(s_t = s|\pi)$ over the state space $S$. Let $d_{t,\pi}(s, a) = p(s_t = s, a_t = a|\pi)$ be its $t$-step state-action counterpart. For the infinite horizon setting, the stationary state distribution is defined as $d_\pi(s) = lim_{t \to \infty} d_{t,\pi}(s)$, and its $\gamma$-discounted version as $d_{\gamma,\pi}(s) = (1 - \gamma) \sum_{t=0}^{\infty} \gamma^t d_{t,\pi}(s)$. We denote the state marginal distribution as $d_{T,\pi}(s) = \frac{1}{T} \sum_{t=0}^{T} d_{t,\pi}(s)$, which is a marginalization of the $t$-step state distribution over a finite time $T$. The objective of maximum entropy exploration is given by:

$$\mathcal{H}(d(\cdot)) = -\mathbb{E}_{s \sim d}[\log(d(s))], \tag{3}$$

where $d$ can be regarded as either the stationary state distribution $d_\pi$ [Mutti and Restelli, 2020], the discounted state distribution $d_{\gamma,\pi}$ [Hazan et al., 2019] or the marginal state distribution $d_{T,\pi}$ [Lee et al., 2019b, Mutti and Restelli, 2020]. In our work we focus on the finite horizon setting and

adapt the marginal state distribution $d_{T,\pi}$ in which $T$ equals the episode horizon $H$, i.e. we seek to maximize the objective:

$$\mathcal{R}_\mathcal{H}(\pi) = \mathbb{E}_{M \sim \hat{P}(M)}\left[\mathcal{H}(d_{H,\pi})\right] = \mathbb{E}_{M \sim \hat{P}(M)}\left[\mathcal{H}\left(\frac{1}{H}\sum_{t=0}^{H} d_{t,\pi}(s)\right)\right], \tag{4}$$

which yields a policy that "equally" visits all states during the episode. Existing works that target maximum entropy exploration rely on estimating the density of the agent's state visitation distribution [Hazan et al., 2019, Lee et al., 2019b]. More recently, a branch of algorithms that employ non-parametric entropy estimation [Liu and Abbeel, 2021a, Mutti et al., 2021, Seo et al., 2021] has emerged, circumventing the burden of density estimation. Here, we follow this common thread and adapt the non-parametric entropy estimation approach; we estimate the entropy using the particle-based $k$-nearest neighbor ($k$-NN estimator) [Beirlant et al., 1997, Singh et al., 2003], as elaborated in the next section.

## 4 The Generalization Ability of Maximum Entropy Exploration

In this section we present an empirical observation—policies trained for maximum entropy exploration (maxEnt policy) generalize well. First, we explain the training procedure of our maxEnt policy, then we show empirical results supporting this observation.

### 4.1 Training State Space Maximum Entropy Policy

To tackle objective (4), we estimate the entropy using the particle-based $k$-NN estimator [Beirlant et al., 1997, Singh et al., 2003], as described here. Let $X$ be a random variable over the support $\chi \subset \mathbb{R}^m$ with a probability mass function $p$. Given the probability of this random variable, its entropy is obtained by $\mathcal{H}_X(p) = -\mathbb{E}_{x \sim p}[\log(p)]$. Without access to its distribution $p$, the entropy can be estimated using $N$ samples $\{x_i\}_{i=1}^N$ by the $k$-NN estimator Singh et al. [2003]:

$$\hat{\mathcal{H}}_X^{k,N}(p) \approx \frac{1}{N}\sum_{i=1}^{N}\log\left(\left\|x_i - x_i^{k\text{-NN}}\right\|_2\right), \tag{5}$$

where $x_i^{k\text{-NN}}$ is the $k$-NN sample of $x_i$ from the set $\{x_i\}_{i=1}^N$.

To estimate the distribution $d_{H,\pi}$ over the states $S$, we consider each trajectory as $H$ samples of states $\{s_t\}_{t=1}^H$ and take $s_t^{k\text{-NN}}$ to be the $k$-NN of the state $s_t$ within the trajectory, as proposed by previous works (APT, Liu and Abbeel [2021a], RE3, Seo et al. [2021], and APS, Liu and Abbeel [2021a]),

$$\hat{\mathcal{H}}^{k,H}(d_{H,\pi}) \approx \frac{1}{H}\sum_{t=1}^{H}\log\left(\left\|s_t - s_t^{k\text{-NN}}\right\|_2\right). \tag{6}$$

Next, similar to previous works, since this sampled estimation of the entropy (Eq. 6) is a sum of functions that operate on each state separately, it can be considered as an expected reward objective $\hat{\mathcal{H}}^{k,H}(d_{H,\pi}) \approx \frac{1}{H}\sum_{t=1}^H r_I(s_t)$ with the intrinsic reward function:

$$r_I(s_t) := \log(\left\|s_t - s_t^{k\text{-NN}}\right\|_2). \tag{7}$$

This formulation enables us to deploy any RL algorithm to approximately optimize objective (4). Specifically, in our work we use the policy gradient algorithm PPO [Schulman et al., 2017], where at every time step $t$ the state $s_t^{k\text{-NN}}$ is chosen from previous states $\{s_i\}_{i=1}^{t-1}$ of the same episode.

Another challenge stems from the computational complexity of calculating the $L_2$ norm of the $k$-NN (Eq. 7) at every time step $t$. To improve computational efficiency, we introduce the following approximation: instead of taking the full observation as the state $s_i$ (i.e. $64 \times 64$ RGB image), we sub-sample (denoted $\downarrow$) the observation by applying average pooling of $3 \times 3$ to produce an image $s_i^\downarrow$ of size $21 \times 21$, resulting in:

$$r_I(s_t) := \log\left(\left\|s_t^\downarrow - s_t^{k\text{-NN},\downarrow}\right\|_2\right). \tag{8}$$

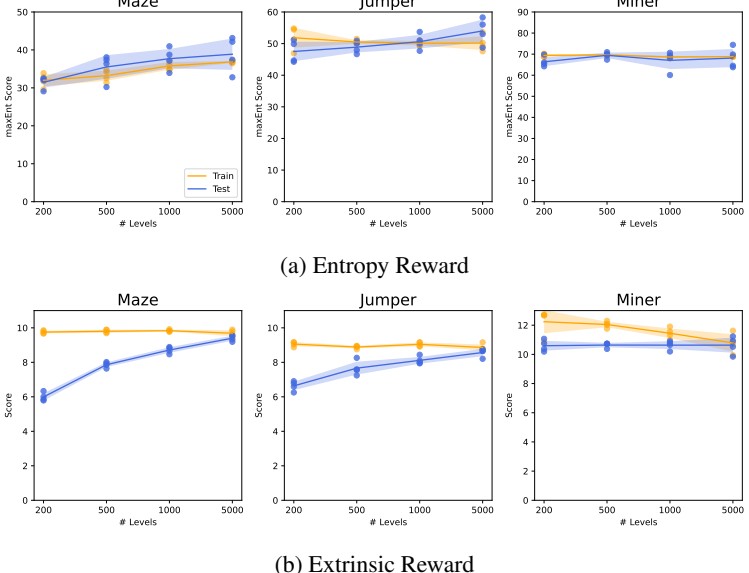

(a) Entropy Reward

(b) Extrinsic Reward

Figure 4: **Generalization ability of maximum entropy vs. extrinsic reward:** (**a**) Score of maximum entropy. (**b**) Score of extrinsic reward. Training for maximum entropy exhibits a small generalization gap in Maze, Jumper and Miner. Average and standard deviation are obtained using 4 seeds.

We emphasize that we do not modify the termination condition of each game. However, a maxEnt policy will learn to *avoid* termination, as this increases the sum of intrinsic rewards. In Figure 3 we display the states visited by a maxEnt policy on Maze. We also experimented with $L_0$ as the state similarity measure instead of $L_2$, which resulted in similar performance (see Appendix F.2).

## 4.2 Generalization of maxEnt Policy

The generalization gap describes the difference between the reward accumulated during training $\mathcal{R}_{emp}(\pi)$ and testing $\mathcal{R}_{pop}(\pi)$ of a policy, where we approximate the population score by testing on a large population of tasks withheld during training. We can evaluate the generalization gap for either an extrinsic reward, or for an intrinsic reward, such as the reward that elicits maxEnt exploration (Eq. 8). In the latter, the generalization gap captures how well the agent's exploration strategy generalizes.

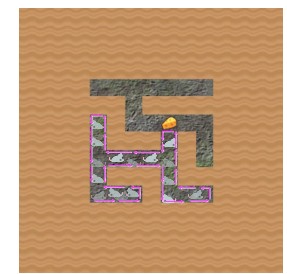

Figure 3: Example of a maxEnt trajectory on Maze. The policy visits every reachable state and averts termination by avoiding the goal state.

We found that agents trained for maximum entropy exploration exhibit a smaller generalization gap compared with the standard approach of training solely with extrinsic reward. Intuitively, this can be attributed to the extrinsic reward serving as an 'easy' signal to learn from, and overfit to in the training environments. To assess the generalization quality of the maxEnt policy, we train agents on $200, 500, 1000$ and $5000$ instances of ProcGen's Maze, Jumper and Miner environments using the intrinsic reward (Eq. 8). The policies are equipped with a memory unit (GRU, Cho et al. 2014) to allow learning of deterministic policies that maximize the entropy [Mutti et al., 2022][3].

The train and test return scores are shown in Fig. 4a. In all three environments, we demonstrate a small generalization gap, as test performance on unseen levels closely follows the performance achieved during training. When considering Maze trained on 200 levels, we observe a small generalization gap of $1.7\%$, meaning test performance closely follows train performance. For Jumper and Miner the maxEnt policy exhibits a small generalization gap of $8.5\%$ and $4.3\%$, respectively. In addition, we verify that the train results are near optimal by comparing with a hand designed approximately optimal exploration policy. For example, on Maze we use the well known maze exploring strategy *wall follower*, also known as the left/right-hand rule [Hendrawan, 2020]; see Appendix B.2 for details.

Next, we evaluate the generalization gap of agents trained to maximize the extrinsic reward[4]. The results for this experiment, shown in Fig. 4b, illustrate that the generalization gap for extrinsic reward

---

[3]An extensive discussion on the importance of memory for the maxEnt objective is in Appendix B.3.

[4]We train for extrinsic reward using an architecture identical to that of the intrinsic reward, with the exception of the memory unit. Incorporating a memory unit in this case further degrades performance (see Appendix B.3).

is more prominent. For comparison, when trained on 200 levels, the figure shows a large generalization gap for Maze (38.8%) and Jumper (27.5%), while Miner exhibits a moderate generalization gap of 13.1%. For an evaluation on all ProcGen games, please see Appendix B.1.

# 5 Explore to Generalize (ExpGen)

Our main insight is that, given the generalization property of the entropy maximization policy established above, an agent can apply this behavior in a test MDP and expect effective exploration *at test time*. In the following, we pair this insight with the epistemic POMDP idea, and propose to play the exploration policy when the agent faces epistemic uncertainty, hopefully driving the agent to a different state where the reward-seeking policy is more certain. This can be seen as an adaptation of the seminal *explicit explore or exploit* idea [Kearns and Singh, 2002], to the setting of ZSG-RL.

## 5.1 Algorithm

Our framework comprises two parts: an entropy maximizing network and an ensemble of networks that maximize an extrinsic reward to evaluate epistemic uncertainty. The first step entails training a network equipped with a memory unit to obtain a maxEnt policy $\pi_{\mathcal{H}}$ that maximizes entropy, as described in section 4.1. Next, we train an ensemble of memory-less policy networks $\{\pi_r^j\}_{j=1}^m$ to maximize extrinsic reward. Following Ghosh et al. [2021], we shall use the ensemble to assess epistemic uncertainty. Different from Ghosh et al. [2021], however, we do not change the RL loss function, and use an off-the-shelf RL algorithm (such as PPO [Schulman et al., 2017] or IDAAC [Stooke et al., 2021]).

At test time, we couple these two components into a combined agent $\pi$ (detailed as pseudo-code in Algorithm 1). We consider domains with a finite action space, and say that the policy $\pi_r^i$ is certain at state $s$ if its action $a_i \sim \pi_r^i(a|s)$ is in consensus with the ensemble: $a_i = a_j$ for the majority of $k$ out of $m$, where $k$ is a hyperparameter of our algorithm. When the networks $\{\pi_r^j\}_{j=1}^m$ are not in consensus, the agent $\pi$ takes a sequence of $n_{\pi_{\mathcal{H}}}$ actions from the entropy maximization policy $\pi_{\mathcal{H}}$, which encourages exploratory behavior.

---

**Algorithm 1** Explore to Generalize (ExpGen)

1: **Input:** ensemble size $m$,
2:          initial state $s_0 = \text{ENVIRONMENT.reset}()$.
3:          $n_{\pi_{\mathcal{H}}} = 0$
4: Train maxEnt policy $\pi_{\mathcal{H}}$ using intrinsic reward $r_I$ (Eq: 7).
5: Train $m$ policies $\pi_r^1, \pi_r^2 \ldots \pi_r^m$ using extrinsic reward $r_{ext}$.
6: **for** $t = 1$ **to** $H$ **do**
7:      $a_i \sim \pi_r^i(\cdot|s_t)$
8:      $n_{\pi_{\mathcal{H}}} \leftarrow n_{\pi_{\mathcal{H}}} - 1$
9:      **if** $a_i \in \text{Consensus}(a_j \mid j \in \{1 \ldots m\})$ and $n_{\pi_{\mathcal{H}}} < 0$ **then**
10:          $a_t = a_i$
11:      **else**
12:          $a_{\mathcal{H}} \sim \pi_{\mathcal{H}}(\cdot|h_t)$
13:          $a_t = a_{\mathcal{H}}$
14:          $n_{\pi_{\mathcal{H}}} \sim Geom(\alpha)$
15:      **end if**
16:      $s_{t+1} \leftarrow \text{ENVIRONMENT.STEP}(a_t)$
17: **end for**

---

**Agent meta-stability** Switching between two policies may result in a case where the agent repeatedly toggles between two states—if, say, the maxEnt policy takes the agent from state $s_1$ to a state $s_2$, where the ensemble agrees on an action that again moves to state $s_1$. To avoid such "meta-stable" behavior, we randomly choose the number of maxEnt steps $n_{\pi_{\mathcal{H}}}$ from a Geometric distribution, $n_{\pi_{\mathcal{H}}} \sim Geom(\alpha)$.

# 6 Experiments

We evaluate our algorithm on the ProcGen benchmark, which employs a discrete 15-dimensional action space and generates RGB observations of size $64 \times 64 \times 3$. Our experimental setup follows ProcGen's 'easy' configuration, wherein agents are trained on 200 levels for $25M$ steps and subsequently tested on random levels [Cobbe et al., 2020]. All agents are implemented using the IMPALA convolutional architecture [Espeholt et al., 2018], and trained using PPO [Schulman et al., 2017] or

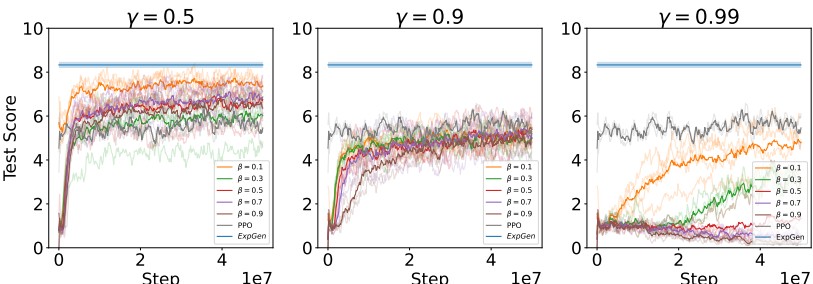

Figure 5: Test performance of PPO trained using the reward $r_{total}$ that combines intrinsic and extrinsic rewards, weighted by $\beta$ (Eq. 9). Each figure details the results for different values of discount factor $\gamma$. All networks are randomly initialized and trained on 200 maze levels, and their mean is computed over 4 runs with different seeds. The figures show an improvement over the PPO baseline for $\gamma = 0.5$. In all cases, ExpGen outperforms the combined reward agent.

IDAAC [Raileanu and Fergus, 2021]. For the maximum entropy agent $\pi_\mathcal{H}$ we incorporate a single GRU [Cho et al., 2014] at the final embedding of the IMPALA convolutional architecture. For all games, we use the same parameter $\alpha = 0.5$ of the Geometric distribution and form an ensemble of 10 networks. For further information regarding our experimental setup and specific hyperparameters, please refer to Appendix C.

## 6.1 Generalization Performance

We compare our algorithm to six leading algorithms: vanilla PPO [Schulman et al., 2017], PLR [Jiang et al., 2021] that utilizes automatic curriculum-based learning, UCB-DrAC [Raileanu et al., 2021], which incorporates data augmentation to learn policies invariant to different input transformations, PPG [Cobbe et al., 2021], which decouples the optimization of policy and value function during learning, and IDAAC [Raileanu and Fergus, 2021], the previous state-of-the-art algorithm on ProcGen that decouples policy learning from value function learning and employs adversarial loss to enforce invariance to spurious features. Lastly, we evaluate our algorithm against LEEP [Ghosh et al., 2021], the only algorithm that, to our knowledge, managed to improve upon the performance of vanilla PPO on Maze and Heist. The evaluation matches the train and test setting detailed by the contending algorithms and their performance is provided as reported by their authors. For evaluating LEEP and IDAAC, we use the original implementation provided by the authors. [5]

Tables 2 and 1 show the train and test scores, respectively, for all ProcGen games. The tables show that ExpGen combined with PPO achieves a notable gain over the baselines on Maze, Heist and Jumper, while on other games, invariance-based approaches perform better (for example, IDAAC leads on BigFish, Plunder and Climber, whereas PPG leads on CaveFlyer, and UCB-DrAC leads on Dodgeball). These results correspond to our observation that for some domains, invariance cannot be used to completely resolve epistemic uncertainty. We emphasize that *ExpGen substantially outperforms LEEP on all games*, showing that our improved exploration at test time is significant. In Appendix E we compare ExpGen with LEEP trained for $50M$ environment steps, showing a similar trend. When combining ExpGen with the leading invariance-based approach, IDAAC, we establish that ExpGen is in-fact *complementary* to the advantage of such algorithms, setting a new state-of-the-art performance in ProcGen. A notable exception is Dodgeball, where all current methods still fail.

Figures 6 and 7 show aggregate statistics of ExpGen, PPO, PLR, UCB-DrAC, PPG and IDAAC for all games [6], affirming the dominance of ExpGen+IDAAC as the state-of-the-art. The results are obtained using 10 runs per game, with scores normalized as in Appendix C.1. The shaded regions indicate $95\%$ Confidence Intervals (CIs) and are estimated using the percentile stratified bootstrap with $2,000$ (Fig. 6) and $50,000$ (Fig. 7) bootstrap re-samples. Fig. 6 (Left) compares algorithm score-distribution, illustrating the advantage of the proposed approach across all games. Fig. 6 (Right)

---

[5]For LEEP and IDAAC, we followed the prescribed hyperparameter values of the papers' authors (we directly corresponded with them). For some domains, we could not reproduce the exact results, and in those cases, we used their reported scores, giving them an advantage.

[6]See *rliable* [Agarwal et al., 2021] for additional details on the various performance measures and protocols.

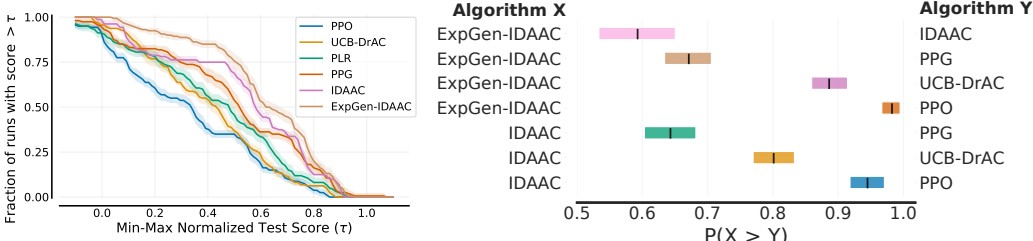

Figure 6: Comparison across all ProcGen games, with $95\%$ bootstrap CIs highlighted in color. **Left.** Score distributions of ExpGen, PPO, PLR, UCB-DrAC, PPG and IDAAC. **Right.** Shows in each row, the probability of algorithm $X$ outperforming algorithm $Y$. The comparison illustrates the superiority of ExpGen over the leading contender IDAAC with probability $0.6$, as well as over other methods with even higher probability.

shows the probability of improvement of algorithm $X$ against algorithm $Y$. The first row (ExpGen vs. IDAAC) demonstrates that the proposed approach surpasses IDAAC with probability $0.6$ and subsequent rows emphasize the superiority of ExpGen over contending methods at an even higher probability. This is because ExpGen improves upon IDAAC in several key challenging tasks and is on-par in the rest. Fig. 7 provides aggregate metrics of mean, median and IQM scores and optimality gap (as $1 - mean$) for all ProcGen games. The figure shows that ExpGen outperforms the contending methods in all measures.

**Ablation Study**    One may wonder if the ensemble in ExpGen is necessary, or whether the observation that the maxEnt policy generalizes well can be exploited using a single policy. We investigate the effect of combining the intrinsic and extrinsic rewards, $r_I$ and $r_{ext}$, respectively, into a single reward as a weighted sum:

$$r_{\text{total}} = \beta r_I + (1 - \beta)r_{\text{ext}}, \tag{9}$$

and train for $\beta = \{0.1, 0.3, 0.5, 0.7, 0.9\}$ on Maze. Figure 5 shows the train and test scores over $50M$ steps for different values of discount factor $\gamma$. We obtain the best test score for $\gamma = 0.5$ and $\beta = 0.1$, illustrating an improvement compared with the PPO baseline. When comparing with ExpGen, the combined reward (Eq. 9) exhibits inferior performance with slightly higher variance. In Appendix C.2 and F, we also provide an ablation study of ensemble size and draw comparisons to variants of our algorithm.

| Game | PPO | PLR | UCB-DrAC | PPG | IDAAC | LEEP | ExpGen (PPO) | ExpGen (IDAAC) |
|---|---|---|---|---|---|---|---|---|
| BigFish | $2.9 \pm 1.1$ | $10.9 \pm 2.8$ | $9.2 \pm 2.0$ | $11.2 \pm 1.4$ | $\mathbf{18.5 \pm 1.2}$ | $4.9 \pm 0.9$ | $6.0 \pm 0.5$ | $\mathbf{18.5 \pm 1.9}$ |
| StarPilot | $24.9 \pm 1.0$ | $27.9 \pm 4.4$ | $30.0 \pm 1.3$ | $\mathbf{47.2 \pm 1.6}$ | $37.0 \pm 2.3$ | $3.2 \pm 2.2$ | $31.0 \pm 0.9$ | $39.8 \pm 2.9$ |
| FruitBot | $26.2 \pm 1.2$ | $28.0 \pm 1.4$ | $27.6 \pm 0.4$ | $27.8 \pm 0.6$ | $\mathbf{27.9 \pm 0.5}$ | $16.4 \pm 1.6$ | $26.2 \pm 0.6$ | $\mathbf{28.4 \pm 0.4}$ |
| BossFight | $7.4 \pm 0.4$ | $8.9 \pm 0.4$ | $7.8 \pm 0.6$ | $\mathbf{10.3 \pm 0.2}$ | $9.8 \pm 0.6$ | $0.5 \pm 0.3$ | $7.7 \pm 0.2$ | $9.8 \pm 0.5$ |
| Ninja | $6.1 \pm 0.2$ | $\mathbf{7.2 \pm 0.4}$ | $6.6 \pm 0.4$ | $6.6 \pm 0.1$ | $6.8 \pm 0.4$ | $4.4 \pm 0.5$ | $6.6 \pm 0.2$ | $6.6 \pm 0.3$ |
| Plunder | $7.8 \pm 1.6$ | $8.7 \pm 2.2$ | $8.3 \pm 1.1$ | $14.3 \pm 2.0$ | $\mathbf{23.3 \pm 1.4}$ | $4.4 \pm 0.3$ | $5.5 \pm 1.3$ | $\mathbf{23.6 \pm 1.4}$ |
| CaveFlyer | $5.5 \pm 0.5$ | $6.3 \pm 0.5$ | $5.0 \pm 0.8$ | $\mathbf{7.0 \pm 0.4}$ | $5.0 \pm 0.6$ | $4.9 \pm 0.2$ | $5.7 \pm 0.3$ | $5.3 \pm 0.7$ |
| CoinRun | $8.6 \pm 0.6$ | $8.8 \pm 0.5$ | $8.6 \pm 0.2$ | $8.9 \pm 0.1$ | $\mathbf{9.4 \pm 0.1}$ | $7.3 \pm 0.4$ | $8.8 \pm 0.1$ | $\mathbf{9.3 \pm 0.3}$ |
| Jumper | $5.8 \pm 0.3$ | $5.8 \pm 0.5$ | $6.2 \pm 0.3$ | $5.9 \pm 0.1$ | $6.3 \pm 0.2$ | $5.4 \pm 1.2$ | $\mathbf{6.7 \pm 0.3}$ | $6.8 \pm 0.5$ |
| Chaser | $3.1 \pm 0.9$ | $6.9 \pm 1.2$ | $6.3 \pm 0.6$ | $\mathbf{9.8 \pm 0.5}$ | $6.8 \pm 1.0$ | $3.0 \pm 0.1$ | $3.6 \pm 1.6$ | $7.1 \pm 1.4$ |
| Climber | $5.4 \pm 0.5$ | $6.3 \pm 0.8$ | $6.3 \pm 0.6$ | $2.8 \pm 0.4$ | $8.3 \pm 0.4$ | $2.6 \pm 0.9$ | $5.9 \pm 0.5$ | $\mathbf{9.5 \pm 0.3}$ |
| Dodgeball | $2.2 \pm 0.4$ | $1.8 \pm 0.5$ | $\mathbf{4.2 \pm 0.9}$ | $2.3 \pm 0.3$ | $3.2 \pm 0.3$ | $1.9 \pm 0.2$ | $2.9 \pm 0.3$ | $2.8 \pm 0.2$ |
| Heist | $2.4 \pm 0.5$ | $2.9 \pm 0.5$ | $3.5 \pm 0.4$ | $2.8 \pm 0.4$ | $3.5 \pm 0.2$ | $4.5 \pm 0.3$ | $\mathbf{7.4 \pm 0.2}$ | $\mathbf{7.2 \pm 0.5}$ |
| Leaper | $4.9 \pm 2.2$ | $6.8 \pm 1.2$ | $4.8 \pm 0.9$ | $\mathbf{8.5 \pm 1.0}$ | $7.7 \pm 1.0$ | $4.4 \pm 0.2$ | $4.0 \pm 1.7$ | $7.6 \pm 1.2$ |
| Maze | $5.6 \pm 0.1$ | $5.5 \pm 0.8$ | $6.3 \pm 0.1$ | $5.1 \pm 0.3$ | $5.6 \pm 0.3$ | $6.6 \pm 0.2$ | $\mathbf{8.3 \pm 0.2}$ | $7.8 \pm 0.2$ |
| Miner | $7.8 \pm 0.3$ | $9.6 \pm 0.6$ | $9.2 \pm 0.6$ | $7.4 \pm 0.2$ | $\mathbf{9.5 \pm 0.4}$ | $1.1 \pm 0.1$ | $8.0 \pm 0.7$ | $\mathbf{9.8 \pm 0.3}$ |

Table 1: **Test score** of ProcGen games trained on 200 levels for $25M$ environment steps. We compare our algorithm to PPO, PLR, UCB-DrAC, PPG, IDAAC and LEEP. The mean and standard deviation are computed over 10 runs with different seeds.

| Game | PPO | PLR | UCB-DrAC | PPG | IDAAC | LEEP | ExpGen (PPO) | ExpGen (IDAAC) |
|---|---|---|---|---|---|---|---|---|
| BigFish | $8.9 \pm 2.0$ | $7.8 \pm 1.0$ | $12.8 \pm 1.8$ | $19.9 \pm 1.7$ | $\mathbf{21.8 \pm 1.8}$ | $8.9 \pm 0.9$ | $7.0 \pm 0.4$ | $\mathbf{21.5 \pm 2.3}$ |
| StarPilot | $29.0 \pm 1.1$ | $2.6 \pm 0.3$ | $33.1 \pm 1.3$ | $\mathbf{49.6 \pm 2.1}$ | $38.6 \pm 2.2$ | $5.3 \pm 0.3$ | $34.3 \pm 1.6$ | $40.0 \pm 2.7$ |
| FruitBot | $28.8 \pm 0.6$ | $15.9 \pm 1.3$ | $29.3 \pm 0.5$ | $\mathbf{31.1 \pm 0.5}$ | $29.1 \pm 0.7$ | $17.4 \pm 0.7$ | $28.9 \pm 0.6$ | $29.5 \pm 0.5$ |
| BossFight | $8.0 \pm 0.4$ | $8.7 \pm 0.7$ | $8.1 \pm 0.4$ | $\mathbf{11.1 \pm 0.1}$ | $10.4 \pm 0.4$ | $0.3 \pm 0.1$ | $7.9 \pm 0.6$ | $9.9 \pm 0.7$ |
| Ninja | $7.3 \pm 0.2$ | $5.4 \pm 0.5$ | $8.0 \pm 0.4$ | $8.9 \pm 0.2$ | $\mathbf{8.9 \pm 0.3}$ | $4.6 \pm 0.2$ | $8.5 \pm 0.3$ | $7.9 \pm 0.6$ |
| Plunder | $9.4 \pm 1.7$ | $4.1 \pm 1.3$ | $10.2 \pm 1.8$ | $16.4 \pm 1.9$ | $24.6 \pm 1.6$ | $4.9 \pm 0.2$ | $5.8 \pm 1.4$ | $\mathbf{26.1 \pm 2.7}$ |
| CaveFlyer | $7.3 \pm 0.7$ | $6.4 \pm 0.1$ | $5.8 \pm 0.9$ | $\mathbf{9.5 \pm 0.2}$ | $6.2 \pm 0.6$ | $4.9 \pm 0.3$ | $6.8 \pm 0.4$ | $5.5 \pm 0.5$ |
| CoinRun | $9.4 \pm 0.3$ | $5.4 \pm 0.4$ | $9.4 \pm 0.2$ | $\mathbf{9.9 \pm 0.0}$ | $9.8 \pm 0.1$ | $6.7 \pm 0.1$ | $9.8 \pm 0.1$ | $9.1 \pm 0.4$ |
| Jumper | $8.6 \pm 0.1$ | $3.6 \pm 0.5$ | $8.2 \pm 0.1$ | $8.7 \pm 0.1$ | $\mathbf{8.7 \pm 0.2}$ | $5.7 \pm 0.1$ | $7.9 \pm 0.2$ | $8.1 \pm 0.4$ |
| Chaser | $3.7 \pm 1.2$ | $6.3 \pm 0.7$ | $7.0 \pm 0.6$ | $\mathbf{10.7 \pm 0.4}$ | $7.5 \pm 0.8$ | $2.6 \pm 0.1$ | $4.7 \pm 1.8$ | $6.9 \pm 1.1$ |
| Climber | $6.9 \pm 1.0$ | $6.2 \pm 0.8$ | $8.6 \pm 0.6$ | $10.2 \pm 0.2$ | $10.2 \pm 0.7$ | $3.5 \pm 0.3$ | $7.7 \pm 0.4$ | $\mathbf{11.4 \pm 0.2}$ |
| Dodgeball | $6.4 \pm 0.6$ | $2.0 \pm 1.1$ | $\mathbf{7.3 \pm 0.8}$ | $5.5 \pm 0.5$ | $4.9 \pm 0.3$ | $3.3 \pm 0.1$ | $5.8 \pm 0.5$ | $5.3 \pm 0.5$ |
| Heist | $6.1 \pm 0.8$ | $1.2 \pm 0.4$ | $6.2 \pm 0.6$ | $7.4 \pm 0.4$ | $4.5 \pm 0.3$ | $7.1 \pm 0.2$ | $\mathbf{9.4 \pm 0.1}$ | $7.0 \pm 0.6$ |
| Leaper | $5.5 \pm 0.4$ | $6.4 \pm 0.4$ | $5.0 \pm 0.9$ | $\mathbf{9.3 \pm 1.1}$ | $8.3 \pm 0.7$ | $4.3 \pm 2.3$ | $4.3 \pm 2.0$ | $8.1 \pm 0.9$ |
| Maze | $9.1 \pm 0.2$ | $4.1 \pm 0.5$ | $8.5 \pm 0.3$ | $9.0 \pm 0.2$ | $6.4 \pm 0.5$ | $9.4 \pm 0.3$ | $\mathbf{9.6 \pm 0.1}$ | $7.4 \pm 0.4$ |
| Miner | $11.3 \pm 0.3$ | $9.7 \pm 0.4$ | $\mathbf{12.0 \pm 0.3}$ | $11.3 \pm 1.0$ | $11.5 \pm 0.5$ | $1.9 \pm 0.6$ | $9.0 \pm 0.8$ | $11.9 \pm 0.2$ |

Table 2: **Train score** of ProcGen games trained on 200 levels for $25M$ environment steps. We compare our algorithm to PPO, PLR, UCB-DrAC, PPG, IDAAC and LEEP. The mean and standard deviation are computed over 10 runs with different seeds.

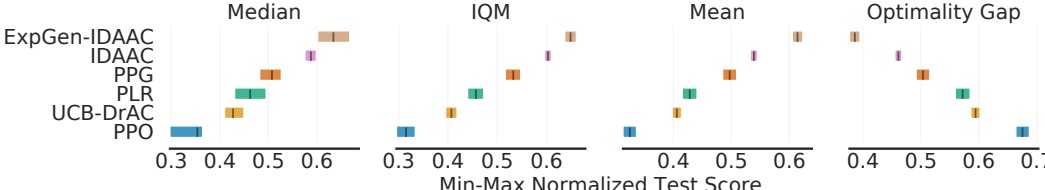

Figure 7: Aggregate metrics for all ProcGen games: mean, median and IQM scores (higher is better) and optimality gap (lower is better), with $95\%$ CIs highlighted in color. ExpGen outperforms the contending methods in all measures.

## 7 Discussion and Limitations

We observed that policies trained to explore, using maximum entropy RL, exhibited generalization of their exploration behavior in the zero-shot RL setting. Based on this insight, we proposed ExpGen—a ZSG-RL algorithm that takes a maxEnt exploration step whenever an ensemble of policies trained for reward maximization does not agree on the current action. We demonstrated that this simple approach performs well on all ZSG-RL domains of the ProcGen benchmark.

One burning question is *why does maxEnt exploration generalize so well?* An intuitive argument is that the maxEnt policy in an MDP is *invariant* to the reward. Thus, if for every training MDP there are many different rewards, each prescribing a different behavior, the maxEnt policy has to be invariant to this variability. In other words, the maxEnt policy contains *no information* about the rewards in the data, and generalization is well known to be bounded by the mutual information between the policy and the training data [Bassily et al., 2018]. Perhaps an even more interesting question is whether the maxEnt policy is also less sensitive to variations in the dynamics of the MDPs. We leave this as an open theoretical problem.

Another consideration is safety. In some domains, a wrong action can lead to a disaster, and in such cases, exploration at test time should be hedged. One possibility is to add to ExpGen's policy ensemble an ensemble of advantage functions, and use it to weigh the action agreement [Rotman et al., 2020]. Intuitively, the ensemble should agree that unsafe actions have a low advantage, and not select them at test time.

Finally, we point out that while our work made significant progress on generalization in several ProcGen games, the performance on Dodgeball remains low for all methods we are aware of. An interesting question is whether performance on Dodgeball can be improved by combining invariance-based techniques (other than IDAAC) with exploration at test time, or whether Dodgeball represents a different class of problems that requires a completely different approach.

**Acknowledgments**    The research of DS was Funded by the European Union (ERC, A-B-C-Deep, 101039436). The research of EZ and AT was Funded by the European Union (ERC, Bayes-RL, 101041250). Views and opinions expressed are however those of the author(s) only and do not necessarily reflect those of the European Union or the European Research Council Executive Agency (ERCEA). Neither the European Union nor the granting authority can be held responsible for them. DS also acknowledges the support of the Schmidt Career Advancement Chair in AI.

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

# A Hidden Maze Experiment

In this section we empirically validate our thought experiment described in the Introduction: we train a recurrent policy on hidden mazes using 128 training levels (with fixed environment colors). Figure 8a depicts the observations of an agent along its trajectory; the agent sees only its own location (green spot), whereas the entire maze layout, corridors and goal, are hidden. In Fig. 8b we visualize the full, unobserved, state of the agent. The train and test results are shown in Fig 9, indicating severe overfitting to the training levels: test performance failed to improve beyond the random initial policy during training. Indeed, the recurrent policy memorizes the agent's training trajectories instead of learning generalized behavior. Hiding the maze's goal and corridors leads to agent-behavior that is the most invariant to instance specific observations—the observations include the minimum information, such that the agent can still learn to solve the maze.

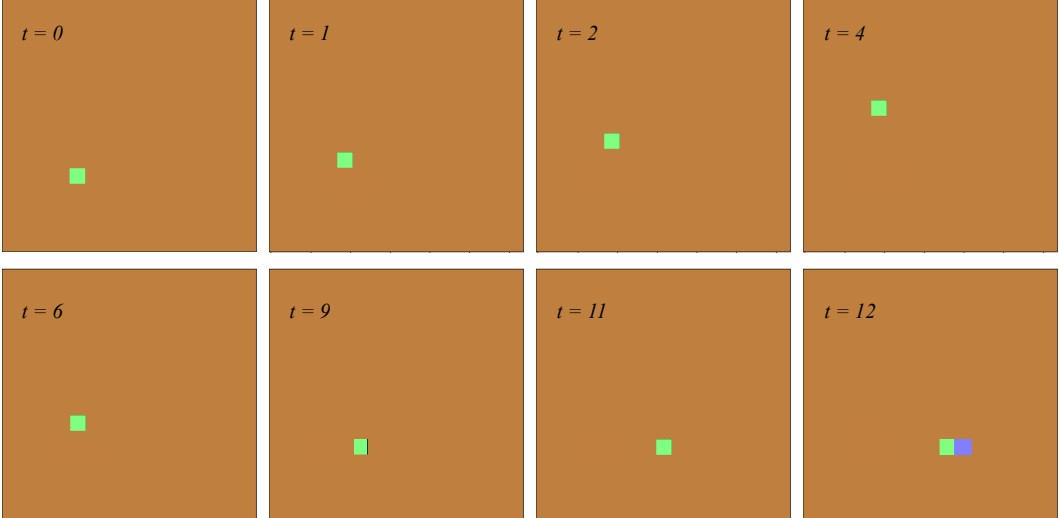

(a) Agent observations along a trajectory for the Hidden Maze task. The bottom-right frame ($t = 12$) shows the agent eventually revealing the goal.

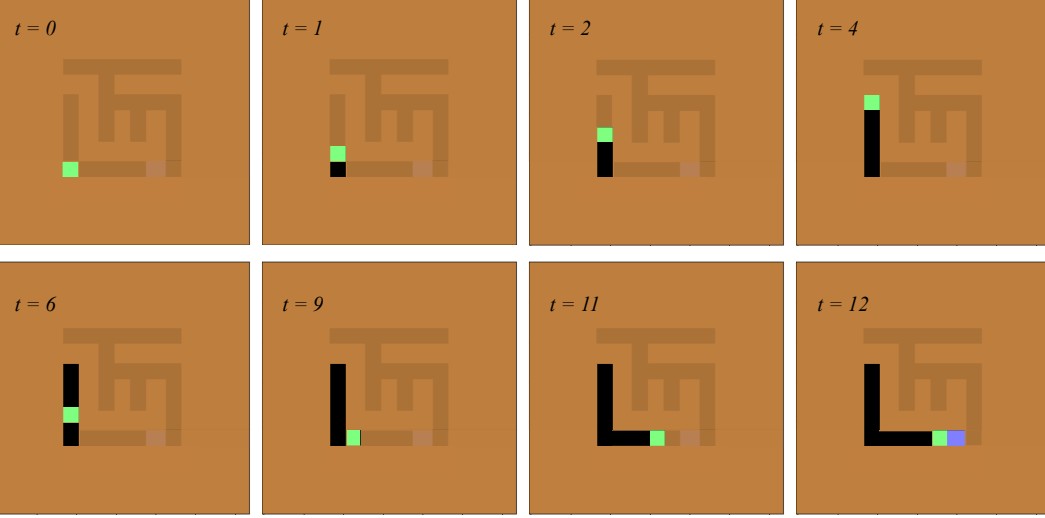

(b) The corresponding *unobserved* true states of agent. Unexplored regions of the maze appear as faded sections.

Figure 8: Hidden Maze experiment where the agent only observes its own location (green spot). Both the goal (purple spot) and corridors are not observable (appear as walls).

This experiment demonstrates why methods based on observation invariance (e.g., IDAAC Raileanu and Fergus [2021]) do not improve performance on Maze-like tasks, despite significantly improving performance on games such as BigFish and Plunder, where invariance to colors helps to generalize.

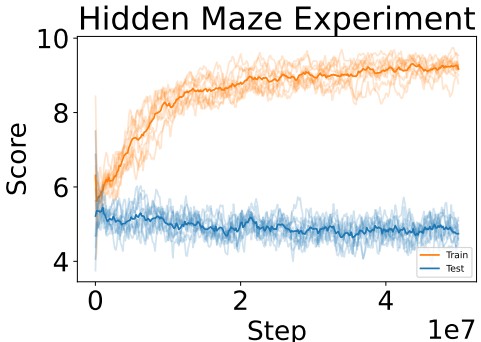

Figure 9: PPO performance on the hidden maze task, indicating severe overfitting. Train (Orange) and Test (blue) performance are displayed for 10 seeds, alongside their means (highlighted in bold).

# B  Maximum Entropy Policy

This section elaborates on the implementation details of the maxEnt oracle and provides a performance evaluation of the maxEnt policy.

## B.1  Generalization Gap of maxEnt vs PPO across all ProcGen Environments

Recall from Section 4.2 that the (normalized) generalization gap is described by

$$(\mathcal{R}_{emp}(\pi) - \mathcal{R}_{pop}(\pi))/\mathcal{R}_{emp}(\pi),$$

where $R_{emp}(\pi) = \mathcal{R}_{train}(\pi)$ and $R_{pop}(\pi) = \mathcal{R}_{test}(\pi)$. Fig 10 shows the generalization ability of the maxEnt exploration policy compared to PPO, obtained from training on 200 training levels.

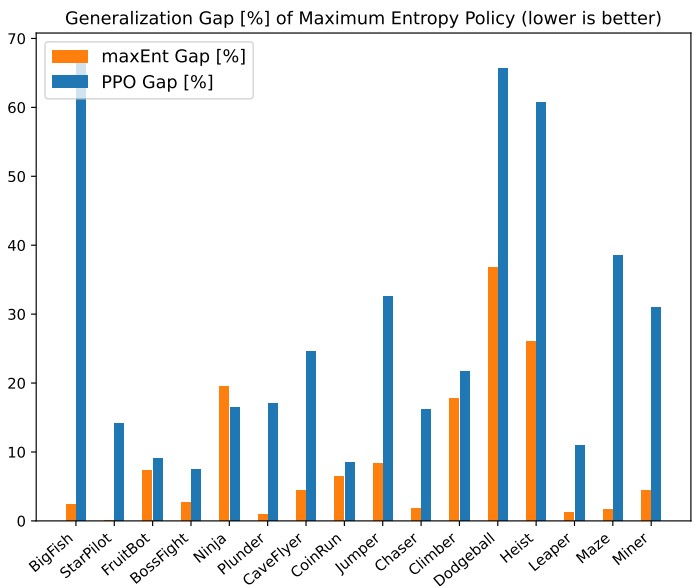

Figure 10: The normalized generalization gap [%] of maxEnt and PPO for all ProcGen games, trained on 200 training levels and averaged over 4 seeds (lower is better).

The figure demonstrates that the maxEnt exploration policies transfer better in zero-shot generalization (achieve a smaller generalization gap) across all ProcGen games apart from Ninja. This holds true

even in environments where the ExpGen algorithm is on par but does not exceed the baseline, pointing to the importance of exploratory behavior in some environments, but not in others.

## B.2 Computing the maxEnt Oracle

The maxEnt oracle prescribes the maximum intrinsic return achievable per environment instance. To simplify the computation of the maxEnt oracle score, in this section, we evaluate the maxEnt score using the $L_0$ instead of the $L_2$ norm and use the first nearest neighbor ($k = 1$). In Maze, we implement the oracle using the right-hand rule [Hendrawan, 2020] for maze exploration: If there is no wall and no goal on the right, the agent turns right, otherwise, it continues straight. If there is a wall (or goal) ahead, it goes left (see Figure 3). In Jumper, we first down-sample (by average pooling) from $64 \times 64$ to $21 \times 21$ pixels to form cells of $3 \times 3$, and the oracle score is the number of background cells (cells that the agent can visit). Here we assume that the agent has a size of $3 \times 3$ pixels. In Miner, the "easy" environment is partitioned into $10 \times 10$ cells. The maximum entropy is the number of cells that contain "dirt" (i.e., cells that the agent can excavate in order to make traversable).

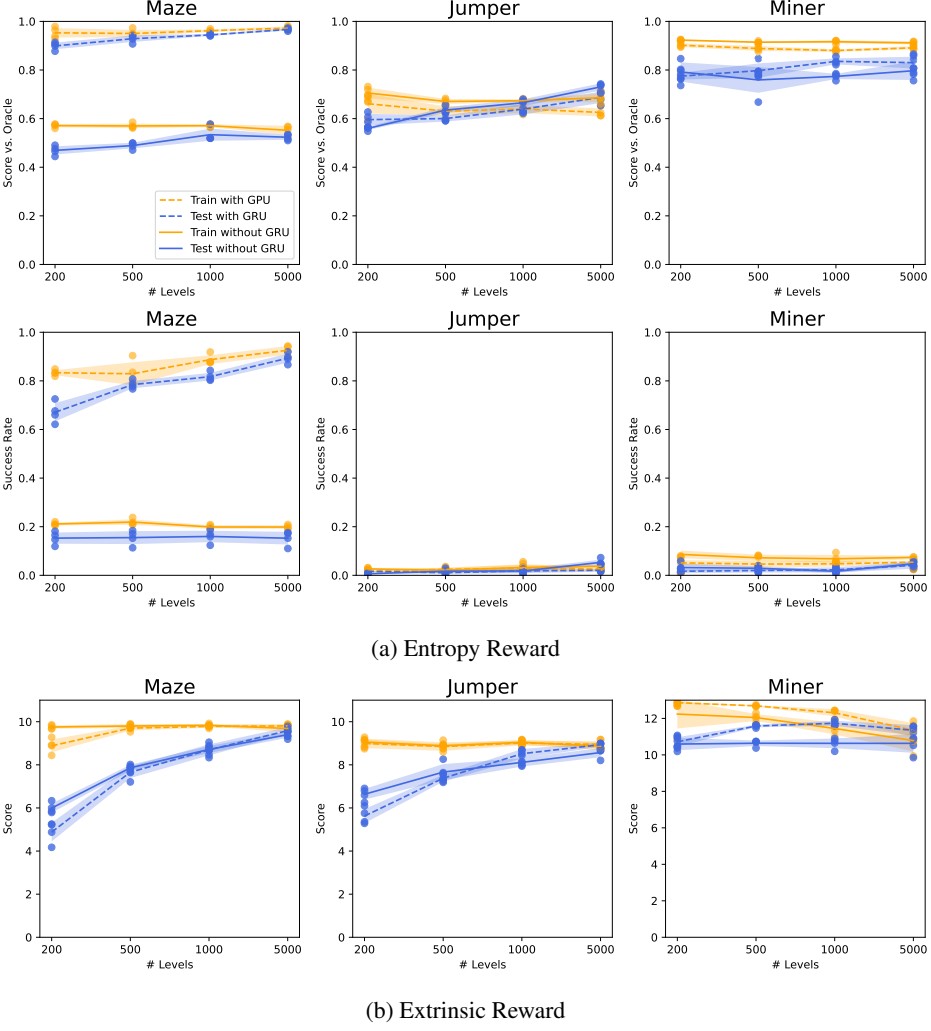

(a) Entropy Reward

(b) Extrinsic Reward

Figure 11: **Generalization ability of maximum entropy and extrinsic reward policy:** (**a.top row**) Score of maximum entropy policy, normalized by the oracle score. (**a.bottom row**) Success Rate of maximum entropy policy. (**b**) Score of extrinsic reward policy. Training for maximum entropy exhibits a small generalization gap in Maze, Jumper and Miner. Average and standard deviation are obtained using 4 seeds.

In Fig. 11a the top row describes the intrinsic return of the maxEnt policy, normalized by the oracle's return (described above). We draw a comparison between agents with memory (GRU) and without. For the Maze environment (top-right), the agent achieves over $90\%$ of the oracle's performance when employing a memory unit (GRU). For the Jumper and Miner environments, the agents approach $60\%$ and $80\%$ of the oracle's score, respectively.

Next, the bottom row of Fig. 11a details the success rate as the ratio of instances in which the agent successfully reached the oracle's score (meaning that the entire traversable region has been explored). For the Maze environment, the agent achieves a success-rate of $70\%$, whereas for the Jumper and Miner the agent fails to meet the oracle's score. This indicates that the oracle, as per our implementation for Jumper and Miner, captures states that are in-effect unreachable, and thus the agent is unable to match the oracle's score.

### B.3 The Importance of a Memory Unit for maxEnt

We evaluate the importance of memory for maxEnt: In Fig. 11a we train a maxEnt agent with and without memory (GRU). The results indicate that for Maze, the GRU is vital in order to maximize performance for all various sizes of training set. In Jumper, we see that the GRU provides an advantage when training on 200 levels, however the benefit diminishes when additional training levels are available. For Miner, there appears to be no significant advantage for incorporating memory with 200 training levels. However, the benefit of a GRU becomes noticeable once more training levels are available (beyond $500$).

When looking at the extrinsic reward (Fig. 11b), we see an interesting effect after introducing a GRU. Maze and Jumper suffer a degradation in performance with 200 levels (indicating overfit), while Miner appears to be unaffected.

In summary, we empirically show that memory is beneficial for the maxEnt policy on Maze, Jumper and Miner. Interestingly, we demonstrate that the introduction of memory for training to maximize extrinsic reward causes the agent to overfit in Maze and Jumper with 200 training levels.

## C Experimental Setup

This section describes the constants and hyperparameters used as part of the evaluation of our algorithm.

### C.1 Normalization Constants

In Figures 2 and 12 we compare the performance of the various algorithms. The results are normalized in accordance with [Cobbe et al., 2020], which defines the normalized return as

$$R_{norm} = (R - R_{min})/(R_{max} - R_{min}).$$

The $R_{min}$ and $R_{max}$ of each environment are detailed in Table 3. Note that the test performance of PPO on Heist (see Fig. 2) is lower than $R_{min}$ (the trivial performance), indicating severe overfitting.

### C.2 Hyperparameters

As described in Section 5.1, the hyperparameters of our algorithm are the number of agents $m$ that form the ensemble, of which $k$ agents are required to be in agreement for the ensemble to achieve a consensus on its action, and $\alpha$ as the parameter of $n_{\pi_{\mathcal{H}}} \sim Geom(\alpha)$ that represents the number of maxEnt steps taken when the ensemble fails to reach a consensus. An additional hyperparameter is the neighborhood size $k_{\mathrm{NN}}$ of the $k$-NN estimator (Section 4.1), used to determine the reward of the maxEnt policy.

We conducted a hyperparameter search over the ensemble size $m \in \{4, 6, 8, 10\}$ for different values of ensemble agreement $k \in \{2, 4, 6, 8\}$, and values of $\alpha \in \{0.2, 0.5, 0.8\}$. We found that a value of $\alpha = 0.5$, and an ensemble size of $m = 10$ produce the best results for all games, whereas the value of $k$ varies from game to game, as detailed in table 4. For example, table 5 shows the results for varying values of $m$ and $k$ for the Maze environment. Throughout our experiments, we train our networks using the Adam optimizer [Kingma and Ba, 2014]. For the PPO hyperparameters we use the hyperparameters found in [Cobbe et al., 2020] as detailed in Table 6.

|  | Hard | | Easy | |
|---|---|---|---|---|
| Environment | $R_{min}$ | $R_{max}$ | $R_{min}$ | $R_{max}$ |
| CoinRun | 5 | 10 | 5 | 10 |
| StarPilot | 1.5 | 35 | 2.5 | 64 |
| CaveFlyer | 2 | 13.4 | 3.5 | 12 |
| Dodgeball | 1.5 | 19 | 1.5 | 19 |
| FruitBot | -.5 | 27.2 | -1.5 | 32.4 |
| Chaser | .5 | 14.2 | .5 | 13 |
| Miner | 1.5 | 20 | 1.5 | 13 |
| Jumper | 1 | 10 | 3 | 10 |
| Leaper | 1.5 | 10 | 3 | 10 |
| Maze | 4 | 10 | 5 | 10 |
| BigFish | 0 | 40 | 1 | 40 |
| Heist | 2 | 10 | 3.5 | 10 |
| Climber | 1 | 12.6 | 2 | 12.6 |
| Plunder | 3 | 30 | 4.5 | 30 |
| Ninja | 2 | 10 | 3.5 | 10 |
| BossFight | .5 | 13 | .5 | 13 |

Table 3: Normalization Constants.

| Game | $k$ |
|---|---|
| Maze | 6 |
| Jumper | 4 |
| Miner | 2 |
| Heist | 8 |
| BigFish | 8 |
| StarPilot | 1 |
| FruitBot | 1 |
| BossFight | 1 |
| Plunder | 2 |
| CaveFlyer | 2 |
| CoinRun | 1 |
| Chaser | 2 |
| Climber | 2 |
| Dodgeball | 2 |
| Leaper | 1 |
| Ninja | 2 |

Table 4: Consensus size $k$ as hyperparameter for each game.

| Ensemble size | 4 | 6 | 8 | 10 |
|---|---|---|---|---|
| ExpGen | $8.02 \pm 0.06$ | $8.15 \pm 0.19$ | $8.00 \pm 0.12$ | $\mathbf{8.22 \pm 0.11}$ |

Table 5: Ablation study of ensemble size and its effect on the test score. Each network in the ensemble is trained on 200 instances of Maze. The results show improved performance for large ensemble size. The mean and standard deviation are computed using 10 runs with different seeds.

Table 7 shows an evaluation of the maxEnt gap and ExpGen using different neighbor size $k_{NN}$ on the Maze environment. We found that the best performance is obtained for $k_{NN} \in \{1, 2, 3, 4, 5\}$. Thus, we choose $k_{NN} = 2$, the second nearest neighbor, for all games.

## D   Results for all ProcGen Games

Figure 12 details the normalized test performance for all ProcGen games. Normalization is performed according to [Cobbe et al., 2020] as described in Appendix C.1. The figure demonstrates

| Parameter | Value |
|---|---|
| $\gamma$ | .999 |
| $\lambda$ | .95 |
| # timesteps per rollout | 512 |
| Epochs per rollout | 3 |
| # minibatches per epoch | 8 |
| Entropy bonus ($k_H$) | .01 |
| PPO clip range | .2 |
| Reward Normalization? | Yes |
| Learning rate | 5e-4 |
| # workers | 1 |
| # environments per worker | 32 |
| Total timesteps | 25M |
| GRU? | Only for maxEnt |
| Frame Stack? | No |

Table 6: PPO Hyperparameters.

| Neighbor Size $k_{\mathrm{NN}}$ | maxEnt (Train) | Maze maxEnt (Test) | Environment maxEnt Gap [%] | ExpGen (Test) |
|---|---|---|---|---|
| 1 | $17.5 \pm 0.5$ | $15.3 \pm 1.4$ | **12.7%** | $7.9 \pm 0.2$ |
| 2 | $33.9 \pm 0.2$ | $31.3 \pm 1.5$ | **7.7%** | $8.3 \pm 0.2$ |
| 3 | $50.2 \pm 2.0$ | $42.5 \pm 2.8$ | **15.3%** | $8.2 \pm 0.2$ |
| 4 | $62.9 \pm 2.6$ | $52.3 \pm 3.0$ | **16.8%** | $8.2 \pm 0.1$ |
| 5 | $70.6 \pm 3.3$ | $57.8 \pm 4.6$ | **15.5%** | $8.2 \pm 0.1$ |
| 6 | $80.0 \pm 3.2$ | $65.3 \pm 1.4$ | 18.4% | $8.1 \pm 0.2$ |
| 7 | $87.2 \pm 1.4$ | $71.1 \pm 1.1$ | 18.5% | $8.0 \pm 0.1$ |
| 8 | $93.8 \pm 1.0$ | $75.3 \pm 2.3$ | 19.7% | $7.8 \pm 0.2$ |
| 9 | $96.6 \pm 4.8$ | $78.0 \pm 2.0$ | 19.3% | $8.1 \pm 0.2$ |
| 10 | $102.2 \pm 1.8$ | $89.1 \pm 0.6$ | **12.8%** | $7.9 \pm 0.1$ |

Table 7: Hyperparameter search for neighborhood size $k_{\mathrm{NN}}$ for the Maze environment. The table presents the maxEnt gap and the performance of ExpGen for varying values of $k_{\mathrm{NN}}$. The mean and variance are computed for 3 seeds.

that ExpGen establishes state-of-the-art results on several challenging games and achieves on-par performance with the leading approach on the remaining games.

# E  Results after convergence

Tables 8 and 9 detail the train and test performance of ExpGen, LEEP and PPO, when trained for $50M$ environment steps. Table 9 shows that ExpGen surpasses LEEP and PPO in most games.

| Game | PPO | LEEP | ExpGen |
|---|---|---|---|
| Maze | $9.45 \pm 0.21$ | $9.84 \pm 0.05$ | $9.71 \pm 0.11$ |
| Heist | $7.97 \pm 0.56$ | $6.86 \pm 0.68$ | $9.62 \pm 0.08$ |
| Jumper | $8.62 \pm 0.08$ | $6.1 \pm 0.4$ | $8.00 \pm 0.13$ |
| Miner | $12.86 \pm 0.06$ | $1.9 \pm 0.2$ | $12.53 \pm 0.12$ |
| BigFish | $14.24 \pm 3.36$ | $8.82 \pm 0.36$ | $5.95 \pm 0.32$ |
| Climber | $8.76 \pm 0.41$ | $4.6 \pm 0.4$ | $8.89 \pm 0.29$ |
| Dodgeball | $8.88 \pm 0.38$ | $6.22 \pm 0.55$ | $8.73 \pm 0.40$ |
| Plunder | $9.64 \pm 1.85$ | $5.1 \pm 0.1$ | $8.06 \pm 0.98$ |
| Ninja | $9.10 \pm 0.32$ | $5.2 \pm 0.3$ | $8.90 \pm 0.30$ |
| CaveFlyer | $8.98 \pm 0.59$ | $5.4 \pm 0.1$ | $8.75 \pm 0.59$ |

Table 8: **Train score** of ProcGen environments trained on 200 instances for 50M environment steps. We compare our algorithm to the baselines LEEP and PPO. The mean and standard deviation are computed over 8 runs with different seeds.

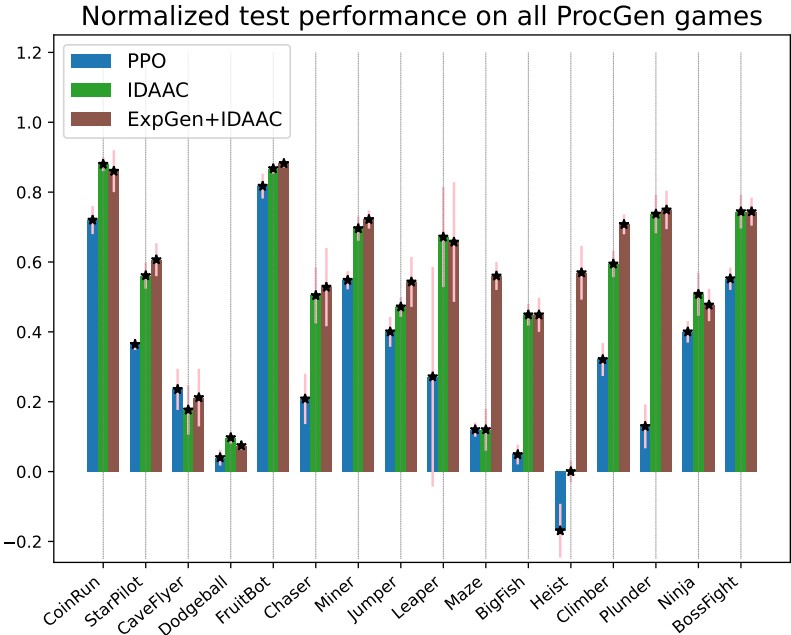

Figure 12: Normalized test Performance for PPO, IDAAC, and ExpGen+IDAAC, on all ProcGen games. ExpGen achieves state-of-the-art performance on test levels of Maze, Heist, and Jumper and on-par performance in the remaining games.

| Game | PPO | LEEP | ExpGen |
|---|---|---|---|
| Maze | $5.78 \pm 0.39$ | $6.78 \pm 0.21$ | $\mathbf{8.33 \pm 0.14}$ |
| Heist | $2.54 \pm 0.45$ | $4.42 \pm 0.57$ | $\mathbf{6.91 \pm 0.24}$ |
| Jumper | $5.78 \pm 0.28$ | $6.4 \pm 0.4$ | $\mathbf{6.64 \pm 0.15}$ |
| Miner | $8.76 \pm 0.33$ | $0.8 \pm 0.1$ | $\mathbf{9.48 \pm 0.39}$ |
| BigFish | $3.82 \pm 1.98$ | $5.5 \pm 0.41$ | $\mathbf{5.99 \pm 0.64}$ |
| Climber | $6.14 \pm 0.50$ | $2.6 \pm 0.4$ | $6.29 \pm 0.54$ |
| Dodgeball | $3.71 \pm 0.55$ | $\mathbf{4.58 \pm 0.47}$ | $3.84 \pm 0.56$ |
| Plunder | $\mathbf{7.78 \pm 1.74}$ | $4.2 \pm 0.2$ | $6.91 \pm 1.00$ |
| Ninja | $6.94 \pm 0.30$ | $4.9 \pm 0.8$ | $\mathbf{6.75 \pm 6.75}$ |
| CaveFlyer | $6.19 \pm 0.66$ | $2.6 \pm 0.2$ | $\mathbf{6.36 \pm 0.49}$ |

Table 9: **Test score** of ProcGen environments trained on 200 instances for $50M$ environment steps. We compare our algorithm to the baselines LEEP and PPO. The mean and standard deviation are computed over 8 runs with different seeds.

## F   Ablation Study

In the following sections, we provide ablation studies of an ExpGen variant that combines random actions and an evaluation of $L_0$ state similarity measure.

### F.1   Ensemble Combined with Random Actions

We compare the proposed approach to a variant of ExpGen denoted by Ensemble+random where we train an ensemble and at test time select a random action if the ensemble networks fail to reach a consensus. The results are shown in Table 10, indicating that selecting the maximum entropy policy upon ensemble disagreements yields superior results.

| Algorithm | Train | Test |
|---|---|---|
| ExpGen | **9.6 ± 0.2** | **8.2 ± 0.1** |
| Ensemble + random | 9.3 ± 0.1 | 6.2 ± 0.2 |
| LEEP | 9.4 ± 0.3 | 6.6 ± 0.2 |
| PPO + GRU | 9.5 ± 0.2 | 5.4 ± 0.3 |
| PPO | 9.1 ± 0.2 | 5.6 ± 0.1 |

Table 10: Ablation study of ExpGen on Maze. The table shows testing scores of networks trained on 200 maze instances. We present a comparison between the proposed approach and LEEP, PPO+GRU and PPO, as well as an alternative ensemble policy with random actions upon ensemble disagreement. The mean and standard deviation are computed using 10 runs with different seeds.

### F.2  Evaluation of Various Similarity Measures for maxEnt

Tables 11 and 12 present the results of our evaluation of ExpGen equipped with a maxEnt exploration policy that uses either the $L_0$ or $L_2$ norms. The experiment targets the Maze and Heist environments and uses the same train and test procedures as in the main paper ($25M$ training steps, score mean and standard deviation are measured over 10 seeds).

| Game | ExpGen $L_0$ (Train) | ExpGen $L_2$ (Train) | PPO (Train) |
|---|---|---|---|
| Heist | **9.4 ± 0.3** | **9.4 ± 0.1** | 6.1 ± 0.8 |
| Maze | **9.6 ± 0.2** | **9.6 ± 0.1** | 9.1 ± 0.2 |

Table 11: Train scores of ExpGen using maxEnt policy with either $L_0$ or $L_2$ compared with PPO. The mean and standard deviation are measured over 10 seeds.

| Game | ExpGen $L_0$ (Test) | ExpGen $L_2$ (Test) | PPO (Test) |
|---|---|---|---|
| Heist | **7.4 ± 0.1** | **7.4 ± 0.2** | 2.4 ± 0.5 |
| Maze | **8.2 ± 0.1** | **8.3 ± 0.2** | 5.6 ± 0.1 |

Table 12: Test scores of ExpGen using maxEnt policy with either $L_0$ or $L_2$ compared with PPO. The mean and standard deviation are measured over 10 seeds.

The results demonstrate that both $L_2$ and $L_0$ allow ExpGen to surpass the PPO baseline for Maze and Heist environments, in which they perform similarly well at test time. This indicates that both are valid measures of state similarity for the maxEnt policy.

## G  Sample Complexity

One may wonder whether the leading approaches would benefit from training on additional environment steps. A trained agent can still fail at test time either due to poor generalization performance (overfitting on a small number of training domains) or due to insufficient training steps of the policy (underfitting). In this work, we are interested in the former and design our experiments such that no method underfits. Figure 13 shows IDAAC training for $100M$ steps on Maze and Jumper, illustrating that the best test performance is obtained at around $25M$ steps, and training for longer does not contribute further (and can even degrade performance). Therefore, although ExpGen requires more environment steps (on account of training its ensemble of constituent reward policies), training for longer does not place our baseline (IDAAC) at any sort of a disadvantage.

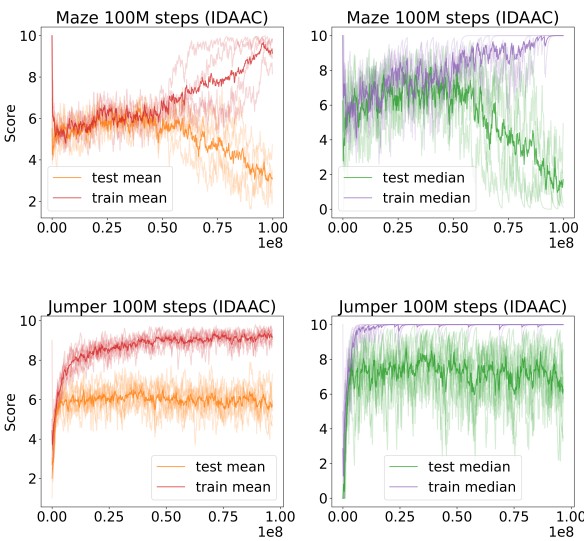

Figure 13: The mean and median of the accumulated reward for IDAAC trained for $100M$ steps, averaged over $10$ runs with different seeds. The curves show that test-reward stagnates and even decreases beyond $25M$ steps.

