# OpenReview forum: "Explore to Generalize in Zero-Shot RL"
_NeurIPS.cc/2023/Conference — NeurIPS 2023 poster_

### Official Review · Reviewer_VwcD · 2023-07-04

**Soundness:** 2 fair
**Presentation:** 2 fair
**Contribution:** 2 fair
**Rating:** 4
**Confidence:** 3

**Summary:**

---
I have raised my score based on the answers and results provided by the authors during the rebuttal.

---
This paper proposes an algorithm named ExpGen that can selectively exhibit a maximum entropy exploration behavior at test time by measuring epistemic uncertainty through ensemble of policies. In order to obtain a policy trained to maximize the entropy, the entropy is formulated as an intrinsic reward, where the sample based approximation to state distribution entropy is obtained using a trajectory of states are used as neighbors. The experimental results show that ExpGen is able to achieve highest score in two of the five ProcGen environments used in the paper, namely, Maze and Heist.

**Strengths:**

The paper proposes novel framework that combines the idea of switching strategies based on the epistemic uncertainty and leveraging the maximum entropy policy to bridge the gap between the performances during train and test time.

**Weaknesses:**

Some parts of the logic are unclear and there are few typos and mistakes in the paper. The experimental draws some what countering argument against the motivation of using the general framework instead of ones based on inductive biases, e.g., IDAAC. This naturally leads to combination of IDAAC and ExpGen, but is mentioned in the paper, but not experimented. Another weakness would be the memory and computational inefficiency, resulting from needing to train and inferencing ensemble of policies. Finally, the empirical studies does not draw a clear picture of why and how ExpGen works, i.e., most results are observed performances of the comprehensive algorithm, lacking a detailed ablation of each components of the algorithm.

**Questions:**

- Why is $k$=1 a good choice (neighbor size)? From the formulation, it seems having just one sample would have a large bias, and thus more samples would lead to better performance.
- The variable $k$ is used twice to represent the majority volume in ensemble. The values seem to largely vary from task to task. Was there a pattern or intuition behind selecting a good candidate for a given task?
- Using difference in RGB images as the difference in states does not quite make sense. Since states within the same trajectory are treated as nearest neighbors, I do not see a clear connection between states that are close in time and close in image space. Shouldn't two have a correlation in order for the entropy estimation to make sense? For example, it would make more sense if the observations are mapped to an embedding space that clusters temporally near states and the embeddings are used to calculate the L2 norm between states.
- Considering the computational burden of training ensemble of policies, calculating L2 distance between two images of size 64x64x3 does not seem like a much of a burden. Was there a large time gap when full original image size was used? How does the performance differ when the original image is used versus different down sampling strategies?
- is $i$ randomly selected?
- Regarding the experiment done in Figure 5, expanding the observed pattern, it seems that lower $\gamma$ and lower $\beta$ may enable PPO to achieve a score similar to ExpGen. Is there an ablation of such?
- While the algorithm is designed upon the idea of preference of generalizing over different tasks (as described  by $\mathcal{L}_{emp}$) compared to inductive biases, the experimental results does not seem to reflect this argument as the performance increase is seen only in a small subset of evaluation tasks.
- As mentioned in 6.1, is there a result of using IDAAC as the base model for ExpGen?

Minor remarks.
- In Equation 5, it is not specified how and where $x_i$ is sampled from.
- Figure 6, legend labels GPU, which should be GRU
- Reference section is missing in the paper (but included in supplementary)

**Limitations:**

Although limitations are included as part of the discussion, some clear limitation could be directly stated, which should help making a clear distinction of ExpGen to other algorithms. For example, comparing memory and computational efficiency against using designs like PPO with extrinsic and intrinsic reward instead of ExpGen with ensemble.

---

> ### Author Rebuttal · Authors · 2023-08-09
>
> We include an evaluation of ExpGen+IDAAC (described in the general comment and accompanied by Rebuttal Figure 1), showcasing state-of-the-art performance on all ProcGen environments.
>
> Regarding memory and computational inefficiencies:
> This is a valid point, which we address in the general comment (accompanied by Rebuttal Figure 2).
>
> Ablation of ExpGen components:
> Thank you for this feedback, we will add ablation experiments to the final version. We also wish to refer you to the appendix (section D) for the ablation study.
>
> Regarding the choice of the value of k in KNN, we conducted hyperparameter tuning for various values - based on which k=1 yielded the best performance. We will include the evaluation in the final version of this paper.
>
> The majority volume of the ensemble k:
> We treat k as a hyperparameter that we tune. Please see Table 10 in the appendix (section D) for an example of tuning k for the maze environment based on an evaluation for various values of k.
>
> In this work we implemented KNN from the RGB image as a way to discern novel states. This was achieved without learned embeddings, but we agree with you that encoding temporally-near states in embedding space has the potential to further improve entropy-maximization performance.
>
> With regards to down-sampling the observed state (full image), our implementation of KNN uses down-sampling primarily in order to smooth out noisy pixels in the observation, as well as extract a more concise state representation that is less sensitive to minor changes. It also helped reduce the training time of the entropy-maximizing policy (MaxEnt) by 50\%.
>
> Regarding an evaluation of lower values of alpha and beta:
> Figure 5 illustrates that even the best values are still significantly worse compared to ExpGen. That said, we will include an evaluation of lower values of alpha and beta in the final version.
>
> Thank you for your insight on combining ExpGen+IDAAC. We include this experiment in the rebuttal: described in the general comment and accompanied with Rebuttal Figure 1. The ExpGen+IDAAC algorithm achieves state-of-the-art results in high-difficulty environments and yields on-par performance in the remaining environments.

---

> > ### Comment · Reviewer_VwcD · 2023-08-16
> > **Thanks for the answers**
> >
> > Thanks for the authors' answers.
> > The comment have cleared some of my concerns. I do still have just few more additional question.
> > - The authors mentioned k=1 (neighbor size) being the optimal found through hyperparameter search. I still do not quite understand why larger sample size for sample-based approximation (beyond 1) gives worse performance. Was the performance similar or significantly worse for larger values of k? Do authors have an insight to why this is the case?
> > - As pointed out by other reviewers as well, the use of L0 norm achieves high intrinsic rewards to visually distinguished pixels. Looking at the environments that ExpGen has achieved high scores, namely Maze and Heist, those environment are ones where the L0 norm between two arbitrary states in a trajectory is almost always similar unless a rewarding event occurs (e.g., only opening a door or eating a cheese to make them disappear creates distinguished visual difference) whereas in other environments, the pixels are more dynamically changing, even when not aligned with the objective of the environment. This leads me to believe that experiments show only one side of the proposed method - the relationship between the choice of the metric between states and some specific properties of the environment is more closely tied to the performance increase than the core idea of the proposed algorithm. Then, the claim ties back to the need of using more general measure of novelty (e.g., expected temporal distance from more larger k-NN neighborhood) to show that the proposed method generally works across different tasks and that its capability is not tied to some specific properties of the environment.
> >
> > Thanks to the authors again for the time.

---

> > > ### Author Response · Authors · 2023-08-20
> > >
> > > Thank you for your comments and your time.
> > >
> > > **The choice of neighbor size $k$ in k-NN:** We added additional experiments to the general comment (accompanied by Rebuttal Tables 4 and 5) that evaluate the MaxEnt score for the Maze and Heist environments for different neighbor sizes $k$. The results show that a small $k$ achieves the best performance (with similar results for $k=1,2,3$), and performance starts to decrease significantly for $k>3$.
> > > Our insight is that allowing for large sample size in Maze and Heist produces additional intrinsic rewards for exploring already-visited states, which hurts exploration performance.
> > >
> > > In the paper, we set the same hyperparameter $k=1$ for all environments (to avoid over-tuning to each specific environment), but we recognize that tuning this hyperparameter for each game separately would improve results. We will add the environment-specific tuning of all games to the appendix of the final version.
> > >
> > > **The use of L0 norm vs L2 in our experiments:** Thank you for pointing out this valuable insight - we added to the general comment an evaluation of ExpGen using MaxEnt exploration with L2 norm instead of L0 (Rebuttal Tables 2 and 3) and we will add this evaluation to the final version.
> > >
> > > The additional results for Maze and Heist using L2 distance show that it is not the L0 metric that leads to the improved performance. Moreover, in preliminary investigations on Maze, we also tried a variant that calculates MaxEnt reward using the true state (in maze it is easy to obtain), which yielded similar results. All in all, we believe these results confirm our intuition that the MaxEnt policy (or a reasonable approximation of it using L0/L2) is more difficult to memorize (overfit) than a reward-seeking policy in the ProcGen Benchmark.
> > >
> > > We believe that the above addresses all the remaining concerns you have raised. Please let us know if there are additional concerns.

---

> > > > ### Comment · Reviewer_VwcD · 2023-08-20
> > > >
> > > > Thanks to the authors for all the answers. I think my questions has been answered from the authors' comments and the results during the rebuttal.
> > > > I have changed my score.

---

> > > > > ### Author Response · Authors · 2023-08-20
> > > > >
> > > > > Thank you for your important feedback and for your willingness to increase the score.
> > > > > We worked hard to address every concern that was raised and believe that this research is valuable to the RL community and to the Zero-shot generalization task in particular.
> > > > >
> > > > > In the rebuttal we extended ExpGen to incorporate IDAAC per your suggestion - this produced a comprehensive algorithm that surpasses the state of the art in several challenging environments (and performs on par with SOTA on the rest).
> > > > > We showed evidence that ExpGen benefits from additional memory and computation resources (through its ensemble mechanism) beyond 25M steps - a stage in which most other algorithms cease to improve.
> > > > > Finally, in the general comment, we provided experiments and ablation studies of the individual components of ExpGen, the L0 vs L2 measures, and a survey of various neighbor sizes (for k-NN).
> > > > > These were helpful in drawing a clearer picture of the inner-workings of ExpGen.
> > > > >
> > > > > We appreciate your insights and would be grateful if you would kindly share any reservations you might still have from accepting our paper.

---

### Official Review · Reviewer_W3Xj · 2023-07-05

**Soundness:** 3 good
**Presentation:** 3 good
**Contribution:** 3 good
**Rating:** 7
**Confidence:** 4

**Summary:**

The paper is based upon the key insight, that maxEnt exploratory policies exhibit a much smaller generalization gap than usual reward seeking policies.
Previous work introduced the framework of epistemic POMDPs, where a random action is chosen until the uncertainty of the policy is low enough again (quantified by policy ensemble members that agree or disagree).
The paper improves upon the previous work by using the maxEnt exploratory policy instead of the random policy when policy ensemble members disagree.
This allows to substantially improve generalization performance on ProcGen tasks previous methods fail on (Maze and Heist).

**Strengths:**

- Very well written introduction and related work.
- Well executed idea to improve weaknesses of previous state-of-the-art algorithm for specific generalization problems of the ProcGen suite.
- Interesting and well executed empirical investigation.

**Weaknesses:**

**Major Weaknesses**

- The usage of L0 norm instead of L2 norm in Eq. (8) is not justified, as to the best of our knowledge, the (log of the) k-NN distance does not approximate the entropy under this norm, but only under the euclidean (L2) norm. Therefore, maximising Eq. (8) might not yield a maxEnt policy.
- Evaluation protocol of main experiments is not clear (see questions).

**Minor Weaknesses**

- In line 72 / 73, it is stated that the exploration policy is GUARANTEED to generalize. The paper does not contain a formal proof, but rather an empirical investigation, therefore this claim should be down-toned to reflect this correctly.

**Remarks**

- Please check the citation style again, often \cite is used instead of \citep.
- Use \eqref to reference equations.
- Check notation (e.g. the true state as part of the history in line 128)

**Questions:**

- How do you select the k-NN? Is it also done using the L0 norm?
- What is your hypothesis why the L0 works better?
- How is it justified to use the L0 norm instead of the L2 norm for the k-NN entropy estimator? I.e. the particle based derivation will not work out.
- The main point of the paper is, that maxEnt exploration policies transfer better in zero-shot generalization. This should be true irrespective of the final performance.
The investigation of this property (Fig. 3 and 6) therefore seems limited, as those are the environments the algorithm actually improves most on. I would raise my score, if the same evaluation could be provided for all environments investigated in the main experiments to get a better view of the empirical significance of this finding.
- How are scores of the main experiments in table 1 and 2 computed?

**Limitations:**

The authors openly discuss limitations of their approach for other environments of the ProcGen suite, which are better suited for different algorithms that incorporate invariances into the trained policy.

---

> ### Author Rebuttal · Authors · 2023-08-09
>
> L2/L0: We used L0 with an intuition that pixel changes are what matters for games like ProcGen, where the agents are localized. This is indeed an approximation, but one that we found to work well. We started an experiment with L2, but did not yet get the results. We remark that L2 is also an approximation (yet indeed a more justified one). We will clarify this point in the text.
>
> Regarding lines 72/73: we agree and will tone down this statement
>
> We are thankful for your inclination to raise your score.
> We are working on evaluating the generalization ability of MaxEnt exploration policy on all environments, and expect to finish the experiments during the discussion phase.
>
> Regarding the score in Tables 1 and 2: They present the average cumulative reward achieved on the train and test environments. Note that for the train environments, only 200 seeds (levels) are used, and for test environments, a new seed is sampled at every episode (from a pool of all possible seeds).

---

> > ### Comment · Reviewer_W3Xj · 2023-08-13
> >
> > Thank you for your additional explanations. However, some questions still remain for me:
> >
> > Could you address the unanswered question about how the k-NN is selected? This should also be stated explicitly in the paper.
> >
> > L2/L0: My intuition about the L0 norm - especially in games like maze where the only thing that changes in the observation during a trajectory is the position of the agent - is that it provides a way of counting unique states. This is rather dissimilar to the L2 norm, which quantifies how much states differ, roughly speaking. In this sense I agree that it can be more meaningful in the shown experiments, but require a lot of knowledge about the problem. Therefore, I would urge to make it more explicit that this is an empirical design choice to improve on very specific environments and would not necessarily hold for a different set of environments, e.g. by adding a paragraph heading to line 191, stating something like "Practical Implementation" or similar.
> >
> > Thank you for the additional details on the score in Tables 1 and 2, however I still don't fully grasp how they are obtained and would ask for a full detailed description, which should also be put into the appendix for future readers. My most pressing questions are, how many test seeds / environemnts are used to calculate the average return. Am I right that the reported STD then is over the average test returns for multiple training runs? How are the train results obtained, averaged over the 200 training seeds? Are the reported results for test and train using the final policy after 25M environment steps or any intermediate ones that might be better?
> >
> > I very much look forward to the additional results on the generalization ability of MaxEnt exploration policies, thank you for your effort!

---

> > > ### Author Response · Authors · 2023-08-18
> > >
> > > The selection of $k$ of the k-NN:
> > > The neighbor size k of k-NN was chosen using hyperparameter tuning. The value of k=1 was assigned to all environments because it produced the most balanced performance across all tasks. Having a single value of k prevents over-tuning of unique k's for each task. We recognize that a more granular per-environment choice of neighbor size can potentially increase performance on a per-task basis and we'll add this evaluation for the Maze and Heist environments in the upcoming comment (with all the remaining environments evaluated for varying k added to the appendix).
> > >
> > > The practicality of L0:
> > > We agree with your insight that the L0 norm is more meaningful in the ProcGen benchmark as it simplifies novel-state detection through counting novel states. We will state this more explicitly in the final version under "Practical Implementation details" and will include experiments using L2 norm for comparison in our upcoming comment  (the experiments are still running).
> > >
> > > Clarification on Tables 1 and 2:
> > > * Number of Test seeds/environments used for calculating average return:
> > > We maintain a buffer of 16K steps obtained by randomly sampling a different seed at the start of each episode from the full distribution of all possible seeds. This means that whenever a game (seed) concludes, e.g. agent death or task completion, the next random seed is drawn. We then compute the average return from the returns of all completed episodes in the buffer. Note that this scheme for computing the average return at test-time is used by the other baselines as well.
> > >
> > > * Reported STD:
> > > Your understanding is correct. The STD is computed over the average test returns (as described above) for 10 training runs (10 different random network-weight initializations).
> > >
> > > * Train results:
> > > We use a buffer of 16K steps and sample at random from the 200 environment seeds of the train set and report the average return of all completed episodes.
> > >
> > > * The reported results of the final policy:
> > > The reported results for test and train are obtained using the final policy after 25M environment steps.
> > >
> > > The generalization ability of MaxEnt exploration policies:
> > > We report the generalization ability of MaxEnt exploration in Rebuttal Table 1 (posted in the general comment above). The table indicates that MaxEnt exploration policies achieve a smaller generalization gap compared to PPO across all ProcGen environments apart from Ninja.

---

> > > > ### Author Response · Authors · 2023-08-20
> > > >
> > > > Thank you for your comments and your time.
> > > >
> > > > Continuing our previous correspondence, we added the following experiments to our general comment:
> > > >
> > > > **The selection of the k-NN:** We added additional experiments that evaluate the MaxEnt score for the Maze and Heist environments for different neighbor sizes $k$ (presented by Rebuttal Tables 4 and 5). The results show that the best performance is achieved for small values of $k$ (with similar results for $k=1,2,3$), and starts to decrease significantly for $k>3$. This supports the choice of $k=1$ in the paper.
> > > >
> > > > **The practicality of L0:** We added to the general comment an evaluation of ExpGen using MaxEnt exploration with L2 norm instead of L0 (Rebuttal Tables 2 and 3). The results show that the performance of L2 is fairly similar to L0 for Maze and Heist (with a slight advantage for L0 in Maze-test).
> > > >
> > > > We will add these evaluations to the final version.

---

> > > > > ### Comment · Reviewer_W3Xj · 2023-08-21
> > > > > **Good Rebuttal**
> > > > >
> > > > > Thank you for the additional explanations and experiments.
> > > > >
> > > > > I was too unspecific regarding the k-NN, I was more referring to the distance measure (I guess L0 as well?) which was used to obtain the k-NN. Furthermore, I appreciate the ablation regarding $k$, I think it adds value to the experimental section. Also the comparison between L0 and L2 norm is insightful.
> > > > >
> > > > > Thank you for providing the results in Rebuttal Table 1, this increases the significance of your findings a lot for me.
> > > > >
> > > > > The authors provided a very good rebuttal and clarified all my concerns. I raised my score correspondingly and recommend acceptance.

---

### Official Review · Reviewer_gmPQ · 2023-07-06

**Soundness:** 3 good
**Presentation:** 2 fair
**Contribution:** 3 good
**Rating:** 7
**Confidence:** 4

**Summary:**

This paper studies zero-shot generalization in RL. They first make an interesting observation: that intrinsic novelty-based rewards corresponding to maximum entropy exploration exhibit a smaller generalization gap than extrinsic environment rewards on ProcGen games. This suggests that MaxEnt rewards are in some sense richer and harder to memorize, and the agent is less prone to overfitting to them and more likely to generalize its exploratory behavior to new levels. The MaxEnt reward here is implemented using a kNN based method, similar to ProtoRL of Yarats et al.

Based on this observation, the paper proposes a new algorithm (ExpGen), which trains a MaxEnt exploratory policy (which, as noted above, should generalize its exploratory behavior well), as well as an ensemble of K exploitation policies using the usual extrinsic reward. At test time, if the ensemble agrees on an action (indicating good generalization), then this action is executed. If not (indicating poor generalization), the exploratory policy (which is assumed to generalize well) is executed instead for a certain number of steps, and the process is repeated.

This algorithm is evaluated on the ProcGen benchmark, where it is compared to a number of other published methods (PPO, PLR, UCB-DrAC, PPG, IDAAC, LEEP). On two games (Maze and Heist), it significantly outperforms the other methods. However, it significantly underperforms in several others.

Overall, this paper has several things going for it: the insight that MaxEnt reward has more favorable generalization properties is definitely interesting, and I think the ExpGen algorithm (or some variant of it) has potential. However, the experiments do not (yet) convincingly make a case for this algorithm: while it shows advantages in some games, it significantly underperforms in others and its aggregate performance does not seem favorable. This may be fixable — as the authors note, the invariances induced by other algorithms are orthogonal to the contributions of ExpGen, so I suspect the benefits could be combined. I would suggest combining the architectural modifications and auxiliary losses from IDAAC with ExpGen - if this indeed combines the best of both algorithms, the resulting method would be a lot more convincing. This is discussed in the paper, but not done, and in my opinion needs to be tested.

Second, there are a number of presentation and/or methodological issues which also need to be addressed. Please see my comments in the Weaknesses section.

I think that if all these issues can be addressed, then this would make a strong submission. I think the substance of this paper is good, but it probably requires another revision cycle to be polished enough for publication.

**Post rebuttal update: the authors have addressed my two main concerns. They have shown that ExpGen can be combined with IDAAC to get robust and SOTA results across all ProcGen games, and they have shown that the baselines do not improve when given a larger sample budget. Based on this, and assuming the authors will also improve the presentation as promised, I think this is now a strong submission and have raised my score to a 7 (Accept).**


**Strengths:**

- The paper addresses an important problem - zero-shot generalization in RL is relatively understudied compared to the singleton MDP setting, but important for many realistic settings
- As mentioned above, the insights are original, and the algorithm (also original) follows nicely from them

**Weaknesses:**

- As mentioned above, the experiments in their current form are not convincing enough
- There is potentially an important methodological issue which needs clearing up, namely ExpGen appears to use many more samples than the other algorithms it is compared to due to the use of several policies.
- The presentation could be improved

**Questions:**

My suggestions for improving the paper are as follows:

Major:

- Try to improve the performance of ExpGen by combining it with algorithmic elements from IDAAC or other methods. As mentioned in the paper, these improvements are orthogonal so hopefully it should be possible — however, it’s not sufficient to hypothesize that the two can be combined, this needs to be validated experimentally.
- It may be that ProcGen isn’t the best experimental testbed to showcase the strengths of the proposed algorithm (indeed, several of the games have dense rewards, and exploration bonuses can sometimes actually hurt performance in such settings). An alternative could be MiniHack (https://arxiv.org/abs/2109.13202) — these environments are also procedurally generated, and additionally for the most part have sparse rewards. You can easily replicate the current setup where there is a limited number of seeds at training, and they are also very fast to run.
- Another potential testbed is the Habitat embodied AI environment. That has a limited number of training levels and overfitting/generalization is a severe issue. In particular, Object Navigation is currently not solvable via RL and current SOTA requires large-scale IL using expensive human demonstrations (https://arxiv.org/abs/2204.03514) (in particular, because they perform exploration which the agent also needs to do at test time). I think this could be a good fit for the proposed method, and if you could get it to work there that would be very convincing.
- Tables 1 and 2 are hard to read due to the many environments and methods. I would suggest displaying aggregate metrics using the RLiable library (https://github.com/google-research/rliable). This will also determine if differences are statistically significant.
- I’m concerned the comparison to baselines might not be fair. It is reported that each method trains for 25M steps. However, ExpGen trains not 1 but (K+1) policies. Are each of these trained for 25M steps, or are they trained for 25M / (K+1) steps? To fairly compare to a policy trained for 25M steps, it should be the latter. Please clarify. If each policy was trained for 25M steps, then the baselines should be trained for 25M * (K+1) steps.




Minor:

- The legend in Figure 3 is quite small and hard to find, which can make the reader confused as to what they’re looking at. I would suggest making it bigger and placing it below the subplots, instead of inside the first one.
- The notation in Section 4.1 is confusing: specifically, the Latex rendering of “k-NN” make it look like “k minus NN”. I would suggest using “x^\mathrm{kNN}” or something else that doesn’t use the minus sign.
- Figure 5: it is not clear from the caption which task this is for. Please add this to the caption.
- The references are currently missing from the main paper (although they show up in the version in the supplement), please fix.

**Limitations:**

Yes.

---

> ### Author Rebuttal · Authors · 2023-08-09
>
> Methodological issue - sample size:
> Thank you for raising this point. We address it in the general comment (accompanied with Rebuttal Figure 2).
>
> Combining ExpGen+IDAAC:
> This is a valuable insight, one that is expressed by the other reviewers as well. In the experiments described in the general comment (accompanied by Rebuttal Figure 1) we address this point and show state-of-the-art performance in all ProcGen games.
>
> Minihack/Habitat:
> Thank you for this suggestion. We will assess how to extend our evaluation to these domains. That said, we believe our results on ProcGen, including the new results in the rebuttal, already establish that ExpGen makes important progress in zero-shot generalization.
>
> Regarding Tables 1 and 2, we'll clarify them and also include aggregate metrics in the revised version.

---

> > ### Comment · Reviewer_gmPQ · 2023-08-16
> > **Thanks, raised my score**
> >
> > Thanks for the response and for running the additional experiments. These have addressed both of my main concerns, therefore I have raised my score to 7 and recommend acceptance.

---

### Official Review · Reviewer_LrJL · 2023-07-07

**Soundness:** 2 fair
**Presentation:** 3 good
**Contribution:** 3 good
**Rating:** 4
**Confidence:** 3

**Summary:**

This work studies generalization on unseen similar tasks, in a zero-shot manner, and discusses how invariance based approach to overfitting might not work all the time. The algorithm proposed, called ExpGen, has one part that explores the space, while a ensemble of agents are trained to do the reward optimization, and it claims to achieve sota results on ProcGen.

**Strengths:**

1. The paper is well written, and it introduces the problem statement and the challenges faced by the current methods well.
2. There is a good discussion about the related work as well, which helps place the proposed method in a proper context.
3. The set of experiments and tables are clear and the authors have given proper pointers to architecture choices and hyperparameters.
4. The limitations section is also addressed very well. The work talks about the game of Dodgeball where all the methods suffer and some possible insights on things that could be looked into for this.
5. The idea looks promising and can help in adding a useful contribution to the community.

**Weaknesses:**

1. Adding training progress plots can show how the evaluation scores evolved.
2. Figure 4 is very difficult to read. It can also benefit from the more useful captions.
3. The proposed method does not do well in all the games. Other than the invariance, is there anything else that might be a reason for this? And is there a study on improving on these games?
4. For some baselines, the scores from the respective papers seem to have been used. It is possible the methods missed a proper hyperparameter tuning. Clarifying this in the work  and doing a proper hyperparameter search and tuning for all methods equally will be helpful.

**Questions:**

- Do all methods, including baselines, go through a proper hyperparameter tuning and search phase conducted by the authors?

**Limitations:**

The authors do discuss the limitations in their work.

---

> ### Author Rebuttal · Authors · 2023-08-09
>
> Training progress:
> ExpGen combines already trained exploration driven policy with an ensemble of (trained) reward policies at test-time. Therefore the evaluation targets test environments and does not produce figures of the training progress.
> We will clarify this point, as well as clarify Figure 4 and its caption in the final version.
>
> ExpGen performance:
> In the paper, ExpGen shows a notable benefit in some environments but not in all. We suspected it is due to the limitations of PPO which forms the ensemble of ExpGen. This is validated in our experiment described in the general comment (please see Fig. 1) where we evaluate ExpGen with an IDAAC ensemble, leading to high performance in all environments.
>
> Hyperparameters for baselines: this is a good point. For LEEP, we performed our own hyperparameter search, following the advice of the paper's first author (we directly corresponded with him). For some domains, however, we could not recover the performance reported in the LEEP paper, and for those domains, we used LEEP's reported numbers, giving an advantage to LEEP. For IDAAC, we used the published code, and corresponded with the first author to verify that we are using the best hyperparameters (she conducted extensive tuning). Again, for some domains we could not reproduce her results, and in those cases used IDAAC's reported scores, giving IDAAC an advantage.

---

> > ### Author Response · Authors · 2023-08-20
> >
> > Dear reviewer, we believe our response and new results should have addressed all the concerns you raised. If you still have concerns, we would appreciate a chance to address them before the discussion period ends.

---

> ### Comment · Reviewer_LrJL · 2023-08-21
>
> Thanks to the authors for spending time and efforts on their rebuttal.
> After getting some more context, I could better understand on why and how some of the choices were made. It is also helpful to understand that the authors tried to reach out to the authors of the respective papers whose results they have reported. However, I am a bit hesitant in increasing the score as without seeing the fully revised final version, it is hard to evaluate the quality of the revised clarifications and notes, especially those related to the figures. I would thus keep my score same as before.
> The authors can greatly benefit from a proper full revision of their work and make this a valuable contribution.

---

> > ### Author Response · Authors · 2023-08-21
> >
> > We wish to emphasize the following:
> > * The **ExpGen+IDAAC** variant establishes a new state-of-the-art on ProcGen (added on August 10 in the rebuttal's single-page PDF [link](https://openreview.net/attachment?id=suDDDKyW2F&name=pdf)) as it surpasses the previous SOTA in several challenging games and is on-par with SOTA on the rest.
> >
> > In addition, we addressed the reviewer's list of weaknesses in the rebuttal:
> > * Weakness#1: Lack of training figures - Since ExpGen ensembles already trained networks, it does not produce training plots but is rather evaluated at test-time. The training plot of PPO is detailed in the main paper (Fig. 5) and of IDAAC in the rebuttal's single-page [PDF](https://openreview.net/attachment?id=suDDDKyW2F&name=pdf).
> > * Weakness#2: Figure 4 caption - The figure depicts the MaxEnt exploration policy in action: an agent explores the Maze environment by applying its learned optimal policy of "wall-following strategy" ([wall follower](https://en.wikipedia.org/wiki/Maze-solving_algorithm#Wall_follower)).
> > * Weakness#3: ExpGen (PPO ensemble) does not do well in all games - We produced an ExpGen+IDAAC variant that excels in all ProcGen games. You can see the figure and its description and clarification in the rebuttal's single-page [PDF](https://openreview.net/attachment?id=suDDDKyW2F&name=pdf).
> > * Weakness#4. Hyperparameters - We precisely detailed the procedure of obtaining the hyperparameters for ExpGen and the baseline methods in the rebuttal: This included working together with the authors of the other leading algorithms to obtain their best-performing setup. We also provide comprehensive hyperparameter ablation studies for ExpGen itself, detailed in rebuttal tables 2-3 (choice of L0 vs L2 metric) and rebuttal tables 4-5 (choice of $k$ neighbor size of k-NN).
> >
> > All in all, the reviewer can see all of the figures that were requested, alongside their description, clarifications, and notes, which will be added in the final version.
> > Kindly note that this year the NeurIPS rebuttal instructions do not allow to modify the paper PDF, but only upload 1 PDF page. We chose to use this page to address the main concerns with factual answers detailing additional experiments.
> > As the reviewer notes in their own review - our paper is well written. The same writing standard will be used for revising our paper based on the discussion phase.

---

### Author Rebuttal · Authors · 2023-08-09

Thank you for your valuable insights and suggestions.
This paper is the first to incorporate exploration driven behavior at test-time towards generalization in RL, and in doing so, achieves state-of-the-art performance in environments that are widely regarded as challenging by all the leading algorithms (e.g. Maze, Heist, Jumper). We are confident that the research community would benefit from this work, due to the significance of zero-shot generalization for RL, and would attract further research into improving our understanding of the role of exploration towards generalization.

We first address all reviewers and describe new results based on the reviewers' suggestions.

In our submission, ExpGen forms an ensemble of reward driven PPO agents. On its own, this approach surpasses the state-of-the-art
in several challenging ProcGen environments (e.g. Maze, Heist, Jumper).
Per the reviewers' recommendation, we combine ExpGen+IDAAC to evaluate variant in which IDAAC reward policies comprise the ensemble (rather than PPO) since IDAAC is the SOTA in all remaining ProcGen environments (e.g., Plunder, Miner, BigFish, etc.) - please see rebuttal Fig. 1.
ExpGen+IDAAC outperforms IDAAC alone in several games (still very significant on Heist, Maze, and Jumper) and performs on-par across all others. **We believe these are strong results**. Moreover, this demonstrates that the applicability of ExpGen is not limited to PPO, and that ExpGen can be applied in the future to other, more powerful reward seeking models to infuse them with exploration driven behavior at test time.

Sample complexity: Several reviewers raised the concern that ExpGen benefits from training on more environment steps (because of exploration + ensemble policies). Let us explain why this is not the case: An agent can fail at test time either due to bad generalization performance (overfitting, e.g., due to a small number of training domains), or due to insufficient training steps of the policy (underfitting). In this work we are interested in the former, and design our experiments such that no method underfits. The Rebuttal Figure 2 shows IDAAC training for 100M steps, which demonstrates that the best test performance is obtained at around 25M steps, and training for longer does not help (and can even degrade performance). Thus, while it is true that our method requires more samples, **our baselines are not in any disadvantage**. Adding constraints on sample complexity to ExpGen is interesting, but is out of scope for this study, which focuses on unlimited samples, but a very limited set of 200 training levels.

---

> ### Author Response · Authors · 2023-08-18
> **Additional experiments on generalization gap of MaxEnt vs PPO across all ProcGen's environments.**
>
> We conducted a full round of experiments evaluating the generalization gap of MaxEnt exploration policy and PPO across all ProcGen's environments -
>
> Rebuttal Table 1:
> | Game         | MaxEnt (Train)     | MaxEnt (Test)      | MaxEnt Gap [%]   | PPO (Train) | PPO (Test) | PPO Gap [%] |
> |--------------|--------------------|--------------------|------------------|------------ | ---------- | ------- |
> | BigFish      | $366.2\pm 12.0$    | $357.6\pm 8.8$     | **2.35%**        | 8.9 ± 2.0   | 2.9 ± 1.1  | 67.4%   |
> | StarPilot    | $440.956\pm 0.04$  | $440.954\pm 0.01$ | **0.00%**        | 29.0 ± 1.1  | 24.9 ± 1.0 | 14.1%   |
> | FruitBot     | $77.782\pm 0.2$    | $72.106\pm 1.3$    | **6.92%**        | 28.8 ± 0.6  | 26.2 ± 1.2 | 8.8%    |
> | BossFight    | $380.576\pm 2.78$ | $370.143\pm 1.44$ | **2.75%**        | 8.0 ± 0.4   | 7.4 ± 0.4  | 7.5%    |
> | Ninja        | $29.356\pm 0.94$   | $23.644\pm 1.27$   | 19.42%           | 7.3 ± 0.2   | 6.1 ± 0.2  | **16.4%** |
> | Plunder      | $353.458\pm 1.7$   | $350.030\pm 0.8$   | **0.98%**        | 9.4 ± 1.7   | 7.8 ± 1.6  | 17.0%   |
> | CaveFlyer    | $52.323\pm 2.6$    | $50.004\pm 2.2$    | **4.43%**        | 7.3 ± 0.7   | 5.5 ± 0.5  | 24.7%   |
> | CoinRun      | $20.47\pm 0.26$    | $19.151\pm 0.06$   | **6.44%**        | 9.4 ± 0.3   | 8.6 ± 0.2  | 8.5%    |
> | Jumper       | $51.85\pm 3.1$     | $47.51\pm 3.1$     | **8.36%**        | 8.6 ± 0.1   | 5.8 ± 0.3  | 32.6%   |
> | Chaser       | $307.707\pm 0.9$   | $302.253\pm 1.6$   | **1.77%**        | 3.7 ± 1.2   | 3.1 ± 0.9  | 16.2%   |
> | Climber      | $21.556\pm 0.46$   | $17.72\pm 0.33$    | **17.65%**       | 6.9 ± 1.0   | 5.4 ± 0.5  | 21.7%   |
> | Dodgeball    | $260.859\pm 3.7$   | $164.785\pm 1.8$   | **36.61%**       | 6.4 ± 0.6   | 2.2 ± 0.4  | 65.6%   |
> | Heist        | $180.910\pm 4.3$   | $133.664\pm 0.97$  | **26.02%**       | 6.1 ± 0.8   | 2.4 ± 0.5  | 60.7%   |
> | Leaper       | $190.782\pm 4.0$   | $188.373\pm 3.5$   | **1.26%**        | 5.5 ± 0.4   | 4.9 ± 2.2  | 10.9%   |
> | Maze         | $31.99\pm 1.6$     | $31.46\pm 1.4$     | **1.65%**        | 9.1 ± 0.2   | 5.6 ± 0.1  | 38.5%   |
> | Miner        | $69.4\pm 0.7$      | $66.36\pm 2.1$     | **4.39%**        | 11.3 ± 0.3  | 7.8 ± 0.3  | 31.0%   |
>
> The table shows the maxEnt and PPO scores, as the cumulative intrinsic/extrinsic reward of the train and test sets respectively. The scores are obtained from training on 200 training seeds. The columns titled MaxEnt Gap and PPO Gap present the (normalized) generalization gap, i.e. $(score(train)-score(test))/score(train)$ for maxEnt and PPO respectively (lower \% is better, with the best performers in bold). The table demonstrates that the MaxEnt exploration policies transfer better in zero-shot generalization (achieve a smaller generalization gap) across all ProcGen games apart from Ninja. This holds even in environments where the ExpGen algorithm does not surpass the baseline, pointing to the importance of exploratory behavior in some environments, but not in others.
>
> Please note that we'll include the experiments on varying values of $k$ (k-NN's neighbor size) and a comparison of the L2 norm in an upcoming comment (the experiments are still running).

---

> > ### Author Response · Authors · 2023-08-20
> >
> > **Evaluation of ExpGen using L2 norm:**
> >
> > The following tables present the result of our evaluation of ExpGen equipped with MaxEnt exploration policy that uses the L2 norm instead of L0. The experiment targets the Maze and Heist environments and uses the same train and test procedure as in the main paper (25M training steps, score average/STD measured over 10 seeds).
> >
> >
> > Train score (Rebuttal Table 2)
> > | Game  | ExpGen L0 (Train)   | ExpGen L2 (Train)   | PPO (Train)   |
> > |-------|---------------------|---------------------|---------------|
> > | Heist | **9.4±0.1**         | 8.1±0.2             | 6.1±0.8       |
> > | Maze  | **9.6±0.1**         | 9.4±0.1             | 9.1±0.2       |
> >
> >
> > Test score (Rebuttal Table 3)
> > | Game  | ExpGen L0 (Test)    | ExpGen L2 (Test)    | PPO (Test)    |
> > |-------|---------------------|---------------------|---------------|
> > | Heist | **7.4±0.2**         | **7.4±0.1**         | 2.4±0.5       |
> > | Maze  | **8.2±0.1**         | 8.0±0.2             | 5.6±0.1       |
> >
> > The results demonstrate that both L0 and L2 allow ExpGen to surpass the PPO baseline for Maze and Heist, where they perform similarly well at test-time. This indicates that both are valid metrics towards the MaxEnt policy (with a slight advantage for L0 in Maze-test).
> >
> > We will include this experiment (extended to all games) in the final version.
> >
> > **Evaluation of MaxEnt using different neighbor size $k$ of the k-NN:**
> >
> > We trained MaxEnt exploration policies using different neighbor sizes ($k=1,2,3,4,5,6,7,8,9$) and L2 norm on the Maze and Heist environments for 25M training steps. Our results are summarized in Tables 4 and 5 for the train and test MaxEnt scores (cumulative intrinsic reward) respectively.
> >
> > Train score of MaxEnt using different neighbor sizes (Rebuttal Table 4)
> > | Game | K=1(Train)   | K=2(Train)   | K=3(Train)   | K=4(Train)    | K=5(Train)    | K=6(Train)    | K=7(Train)    | K=8(Train)  | K=9(Train)  |
> > |-------|---------------------|---------------------|---------------------|---------------------|---------------------|---------------------|---------------------|---------------------|---------------------|
> > | Heist | **202.9±16.7**      | **192.4±10.8**      | **188.5±8.8**       | 181.4±6.5           | 178.1±9.5           | 156.4±9.1           | 175.9±8.8           | 145.2±8.2           | 145.6±2.9         |
> > | Maze  | **76.4±7.8**        | **69.8±9.4**        | **66.8±12.0**       | 65.2±9.0            | 60.4±8.4            | 58.0±10.9           | 52.4±8.3            | 50.5±6.2            | 45.5±4.5            |
> >
> > Test score of MaxEnt using different neighbor sizes (Rebuttal Table 5)
> > | Game  | K=1 (Test)          | K=2 (Test)          | K=3 (Test)          | K=4 (Test)          | K=5 (Test)          | K=6 (Test)          | K=7 (Test)          | K=8 (Test)          | K=9 (Test)          |
> > |-------|---------------------|---------------------|---------------------|---------------------|---------------------|---------------------|---------------------|---------------------|---------------------|
> > | Heist | **190.5±6.9**       | **185.5±4.8**       | 178.3±5.2           | 171.4±9.3           | 164.0±2.7           | 141.8±8.4           | 150.0±2.0           | 124.0±4.5           | 123.2±3.2           |
> > | Maze  | **58.4±1.6**        | **59.4±2.2**        | **57.6±2.1**        | 54.7±0.9            | 48.1±0.2            | 47.6±0.4            | 46.3±2.8            | 42.3±1.3            | 39.1±0.7            |
> >
> > The tables show that training MaxEnt policies using small $k$ ($k=1,2,3$) yields the best test performance on Maze and Heist, and performance starts to decrease dramatically for $k=4$ and above (the drop in performance for $k>3$ exceeds the error standard deviation). In our paper, we based our choice of $k=1$ on these results and applied it to all environments.
> >
> > We will add this evaluation, and extend it to all environments in the final version of the appendix.

---

### Decision · Program_Chairs · 2023-09-21

**Decision:**

Accept (poster)

**Comment:**

This paper proposes a new method for zero-shot generalisation in RL, where the main idea is to train an ensemble of policies using extrinsic rewards along with an exploration policy using an intrinsic reward for maximum state entropy, and use either of them to act at test time, depending on how much the ensemble of policies agree with each other. The majority of the reviewers agreed that the proposed method is novel and sensible. Some of the reviewers had concerns around the poor performances on some tasks and the justification of the choice of distance metric for the kNN method. However, the authors provided much stronger results and conducted additional ablations for kNN during the rebuttal period, and the reviewers were satisfied with the new results. One of the reviewers who expressed a similar concern did not follow up during the discussion period. Since the rest of the reviewers said the new results addressed their concerns, I recommend accepting this paper.